# Parameters as interacting particles: long time convergence and asymptotic error scaling of neural networks

**Grant M. Rotskoff**
Courant Institute of Mathematical Sciences
New York University
rotskoff@cims.nyu.edu

**Eric Vanden-Eijnden**
Courant Institute of Mathematical Sciences
New York University
eve2@cims.nyu.edu

## Abstract

The performance of neural networks on high-dimensional data distributions suggests that it may be possible to parameterize a representation of a *given* high-dimensional function with controllably small errors, potentially outperforming standard interpolation methods. We demonstrate, both theoretically and numerically, that this is indeed the case. We map the parameters of a neural network to a system of particles relaxing with an interaction potential determined by the loss function. We show that in the limit that the number of parameters $n$ is large, the landscape of the mean-squared error becomes convex and the representation error in the function scales as $O(n^{-1})$. In this limit, we prove a dynamical variant of the universal approximation theorem showing that the optimal representation can be attained by stochastic gradient descent, the algorithm ubiquitously used for parameter optimization in machine learning. In the asymptotic regime, we study the fluctuations around the optimal representation and show that they arise at a scale $O(n^{-1})$. These fluctuations in the landscape identify the natural scale for the noise in stochastic gradient descent. Our results apply to both single and multi-layer neural networks, as well as standard kernel methods like radial basis functions.

## 1 Introduction

The methods and models of machine learning are rapidly becoming *de facto* tools for the analysis and interpretation of large data sets. The ability to synthesize and simplify high-dimensional data raises the possibility that neural networks may also find applications as efficient representations of known high-dimensional functions. In fact, these techniques have already been explored in the context of free energy calculations [1], partial differential equations [2, 3], and forcefield parameterization [4]. Yet determining the optimal set of parameters or "training" a given neural network remains one of the central challenges in applications due to the slow dynamics of training [5] and the complexity of the objective function [6, 7]. Parameter optimization in machine learning typically relies on the stochastic gradient descent algorithm (SGD), which makes an empirical estimate of the gradient of the objective function over a small number of sample points [5]. SGD has been analyzed in some cases—for example, when the problem is known to be convex, as in the over-parameterized limit or other idealized settings [8, 9, 10, 11], there are rigorous guarantees of convergence and estimates of convergence rates [12].

While finding the best set of parameters is computationally challenging, we have strong theoretical guarantees that neural networks can represent a large class of functions. The universal approximation theorems [13, 14, 15] ensure the existence of a (possibly large) set of parameters that bring a neural network arbitrarily close to a given function over a compact domain. A similar statement has been

proved for radial basis functions [16]. However, the proofs of the universal approximation theorems do not ensure that any particular optimization technique can locate the ideal set of parameters.

**Parameters as particles**—In order to study the properties of stochastic gradient descent for neural network optimization, we recast the standard training procedure in terms of a system of interacting particles [17]. In doing so, we give an exact rewriting of stochastic gradient descent as a stochastic differential equation with multiplicative noise, which has been studied previously [18, 19]. We interpret the limiting behavior of the parameter optimization via a nonlinear Liouville equation for the time evolution of a parameter distribution [20]. This framework provides analytical tools to determine a Law of Large Numbers for the convergence of the optimization and to derive scaling results for the error term as time and the number of parameters grow large. A similar perspective has been adopted concurrently by Mei et al. [21], Chizat and Bach [22], and Sirignano and Spiliopoulos [23], which study the "mean field limit", similar to our Law of Large Numbers, but not asymptotic fluctuations or error scaling.

**Convergence and asymptotic dynamics of stochastic gradient descent**—We demonstrate that the optimization problem becomes convex in the limit $n \to \infty$ and we show that both gradient descent and SGD convergence to the global minimum [24, 21]. This argument shows that the universal approximation theorem can be obtained as the limit of a stochastic gradient based optimization procedure under an appropriate choice of hyper-parameters. In the scaling limit, our analysis gives bounds on the error of a representation and characterizes the asymptotic fluctuations in that error. Convergence to the optimum to first order occurs rapidly, i.e. on $O(1)$ timescales. Diminishing the error at next order requires quenching the noise in the dynamics on $O(\log n)$ time scales.

**Implications of noise in descent dynamics**—Our results give an explicit theoretical explanation for the observation that additional noise in can lead to better generalization for neural networks [25, 26]; local minima of depth $O(n^{-1})$ are washed out by the noise of SGD.

**Numerical experiments**—We verify the scaling predicted by our asymptotic arguments for single layer neural networks. Because it is impossible to determine the exact interaction potential in general, we carry out numerical experiments using stochastic gradient descent for ReLU neural networks. We use the $p$-spin energy function [27, 28] as the target function due to its complexity as the dimension grows large.

**Key assumptions**—In order to derive the stochastic partial differential equation for SGD, we effectively assume the large data limit. Because we are focusing on function approximation we can always generate new training data by sampling random points in the domain of the function and evaluating the target function at those points. The partial differential equation for gradient descent represents the evolution of the parameters on the true loss landscape, i.e., the large data limit. In this limit, the dynamics is similar to online algorithms for stochastic gradient descent [5].

## 2 Parameters as particles

Given a function $f : \Omega \to \mathbb{R}$ defined on a compact set $\Omega \subset \mathbb{R}^d$, consider its approximation by

$$f_n(\boldsymbol{x}) = \frac{1}{n} \sum_{i=1}^{n} c_i \varphi(\boldsymbol{x}, \boldsymbol{y}_i) \tag{1}$$

where $n \in \mathbb{N}$, $\varphi : \Omega \times D \to \mathbb{R}$ is some kernel and $(c_i, \boldsymbol{y}_i) \in \mathbb{R} \times D$ with $D \subset \mathbb{R}^N$. The $c_i$ and $\boldsymbol{y}_i$ are parameters to be learned for $i = 1, \ldots, n$. We place the following assumption on the kernel: for any test function $h$,

$$\forall \boldsymbol{y} \in D : \int_{\Omega} h(\boldsymbol{x}) \varphi(\boldsymbol{x}, \boldsymbol{y}) d\mu(\boldsymbol{x}) = 0 \quad \Rightarrow \quad h(\boldsymbol{x}) = 0, \quad \forall \boldsymbol{x} \in \Omega, \tag{2}$$

where $\mu$ is some positive measure on $\Omega$ (for example the Lebesgue measure, $d\mu(\boldsymbol{x}) = d\boldsymbol{x}$). This condition is satisfied for nonlinearities typically encountered in machine learning; a neural network with any number of layers using a positive nonlinear activation function (e.g., ReLU, sigmoid) will clearly satisfy this property if the linear coefficients are non-zero. The property above is similar to the discriminatory kernel condition in Cybenko [13]. Our results apply to radial basis functions, single layer neural networks, and multilayer neural networks in which the final layer is scaled with $n$. In particular, the statements we make require a "wide" final layer but are still applicable to networks with multiple layers.

By "training" the representation, we mean that we seek to optimize the parameters so as to minimize the mean-squared error loss function,

$$\ell(f, f_n) = \tfrac{1}{2} \int_\Omega |f(\boldsymbol{x}) - f_n(\boldsymbol{x})|^2 \, d\mu(\boldsymbol{x}). \tag{3}$$

In this case we have chosen to employ the mean-squared error and we can view $\ell(f, f_n)$ as an "energy" function for the parameters $\{(c_i, \boldsymbol{y}_i)\}_{i=1}^n$,

$$E(c_1, \boldsymbol{y}_1, \ldots, c_n, \boldsymbol{y}_n) := n\left(\ell(f, f_n) - C_f\right) = \sum_{i=1}^n c_i F(\boldsymbol{y}_i) + \frac{1}{2n} \sum_{i,j=1}^n c_i c_j K(\boldsymbol{y}_i, \boldsymbol{y}_j) \tag{4}$$

where $C_f = \tfrac{1}{2} \int_\Omega |f(\boldsymbol{x})|^2 \, d\mu(\boldsymbol{x})$ is a constant unaffected by the optimization and we have defined

$$F(\boldsymbol{y}) = \int_\Omega f(\boldsymbol{x})\varphi(\boldsymbol{x}, \boldsymbol{y})d\mu(\boldsymbol{x}) \qquad K(\boldsymbol{y}, \boldsymbol{z}) = \int_\Omega \varphi(\boldsymbol{x}, \boldsymbol{y})\varphi(\boldsymbol{x}, \boldsymbol{z})d\mu(\boldsymbol{x}). \tag{5}$$

Directly optimizing the coefficients to minimize the loss function $\ell$ is challenging in general because we do not have any guarantee of convexity. However, these difficulties can be conceptually alleviated by instead writing the objective function in terms of a weighted distribution

$$G_n : D \to \mathbb{R}, \quad G_n(\boldsymbol{y}) = \frac{1}{n} \sum_{i=1}^n c_i \delta(\boldsymbol{y} - \boldsymbol{y}_i) \tag{6}$$

which converges weakly to some $G(\boldsymbol{y})$ as $n \to \infty$, a fact which we describe in detail below. Convolution with this weighted distribution provides a convenient expression for the function representation

$$f_n(\boldsymbol{x}) = \int_D \frac{1}{n} \sum_{i=1}^n c_i \varphi(\boldsymbol{x}, \boldsymbol{y})\delta(\boldsymbol{y} - \boldsymbol{y}_i)d\boldsymbol{y} \equiv \varphi \star G_n. \tag{7}$$

Interestingly, in the limit that $n \to \infty$ the objective function for the optimization becomes convex in terms of the signed distribution,

$$\ell(f, \varphi \star G) = \tfrac{1}{2} \int_\Omega |f(\boldsymbol{x}) - (\varphi \star G)(\boldsymbol{x})|^2 \, d\mu(\boldsymbol{x}). \tag{8}$$

meaning that a unique minimum value of the loss function can be attained for a not necessarily unique minimizer $G^*$ for which $\ell(f, \varphi \star G^*) = 0$. This observation formalizes the statements made by Bengio et al. in Ref. [24]. While the objective function is convex, it is by no means trivial to optimize the weighted distribution. Writing the loss function in this language gives us a perspective that can be exploited to derive the scaling of the error in arbitrary neural networks trained with stochastic gradient descent.

## 3 Gradient descent

We first discuss the case of gradient descent for which we provide derivations of a law of large numbers (LLN) and central limit theorem (CLT) for the optimization dynamics. These statements reveal the scaling in the representation error and the analysis has synergies which are useful in deriving LLN and CLT for stochastic gradient descent. Detailed arguments for the propositions stated here are provided in the supplementary material.

The gradient descent dynamics is given by coupled ordinary differential equations for the weight and the parameters of the kernel,

$$\begin{cases} \dot{\boldsymbol{Y}}_i = C_i \nabla F(\boldsymbol{Y}_i) - \dfrac{1}{n} \sum_{j=1}^n C_i C_j \nabla K(\boldsymbol{Y}_i, \boldsymbol{Y}_j), \\[2ex] \dot{C}_i = F(\boldsymbol{Y}_i) - \dfrac{1}{n} \sum_{j=1}^n C_j K(\boldsymbol{Y}_i, \boldsymbol{Y}_j) \end{cases} \tag{9}$$

with initial conditions sampled independently from a probability distribution $\rho_{\text{in}}(\boldsymbol{y}, c)$ with full support in the domain $D \times \mathbb{R}$. We analyze the evolution of the parameters by studying the "particle" distribution

$$\rho_n(t, \boldsymbol{y}, c) = \frac{1}{n} \sum_{i=1}^{n} \delta(c - C_i(t)) \delta(\boldsymbol{y} - \boldsymbol{Y}_i(t)) \tag{10}$$

the first moment of which is the weighted distribution (6),

$$G_n(t, \boldsymbol{y}) = \int c\rho_n(t, \boldsymbol{y}, c)dc = \frac{1}{n} \sum_{i=1}^{n} C_i(t)\delta(\boldsymbol{y} - \boldsymbol{Y}_i(t)). \tag{11}$$

We can express the function representation in terms of the distribution as $f_n(t, \boldsymbol{x}) = \int \varphi(\boldsymbol{x}, \boldsymbol{y}) G_n(t, \boldsymbol{y}) d\boldsymbol{y}$. Taking the limit $n \to \infty$, we see that the zeroth order term of the distribution has smooth initial data $\rho_0(0) = \rho_{\text{in}}$ by the Law of Large Numbers. In Sec S1.1 we derive a nonlinear partial differential equation satisfied by $\rho_0$, essentially by applying the chain rule:

$$\partial_t \rho_0 = \nabla \cdot (c\nabla U([\rho_0], \boldsymbol{y})\rho_0) + \partial_c (U([\rho_0], \boldsymbol{y})\rho_0), \tag{12}$$

where

$$U([\rho], \boldsymbol{y}) = -F(\boldsymbol{y}) + \int_{D \times \mathbb{R}} c' K(\boldsymbol{y}, \boldsymbol{y}')\rho(\boldsymbol{y}', c')d\boldsymbol{y}'dc' \tag{13}$$

The PDE (12) is gradient descent in Wasserstein metric on a convex energy functional of the density (cf. Sec. S1.2.1); we refer to this type of equation as a nonlinear Liouville equation.

## 3.1 Law of large numbers

The limiting equation (12) is a well-posed and deterministic nonlinear partial integro-differential equation. We can express it in terms of the target function $f(\boldsymbol{x})$ by denoting

$$f_0(t, \boldsymbol{x}) = \int_{D \times \mathbb{R}} c\varphi(\boldsymbol{x}, \boldsymbol{y})\rho_0(t, \boldsymbol{y}, c)d\boldsymbol{y}dc \tag{14}$$

and we see that

$$\partial_t f_0(t, \boldsymbol{x}) = -\int_{\Omega} M([\rho_0(t)], \boldsymbol{x}, \boldsymbol{x}') (f_0(t, \boldsymbol{x}) - f(\boldsymbol{x})) \, d\mu(\boldsymbol{x}') \tag{15}$$

where the symmetric kernel function $M$ is given by

$$M([\rho], \boldsymbol{x}, \boldsymbol{x}') = \int_{D \times \mathbb{R}} \left(c^2 \nabla_{\boldsymbol{y}} \varphi(\boldsymbol{x}, \boldsymbol{y}) \cdot \nabla_{\boldsymbol{y}} \varphi(\boldsymbol{x}', \boldsymbol{y}) + \varphi(\boldsymbol{x}, \boldsymbol{y})\varphi(\boldsymbol{x}', \boldsymbol{y})\right) \rho(\boldsymbol{y}, c) \, d\boldsymbol{y}dc. \tag{16}$$

This kernel is positive definite and symmetric implying that the only stable fixed point is $f_0 = f$ if $\rho_0(t = 0) = \rho_{\text{in}} > 0$, as discussed in Sec S1.2. Fixed points of the gradient flow that are not energy minimizers exist, but they are not dynamically accessible from the initial density that we use (cf. [22] and Sec S1.2).

**Proposition 3.1 (LLN for gradient descent)** *Let* $f_n(t) = f_n(t, \boldsymbol{x}) = \sum_{i=1}^{n} C_i(t)\varphi(\boldsymbol{x}, \boldsymbol{Y}_i(t))$ *where* $\{\boldsymbol{Y}_i(t), C_i(t)\}_{i=1}^{n}$ *are the solution of* (9) *for the initial condition where each pair* $(\boldsymbol{Y}_i(0), C_i(0))$ *is sampled independently from* $\rho_{\text{in}} > 0$. *Then*

$$\lim_{n \to \infty} f_n(t) = f_0(t) \qquad \mathbb{P}_{\text{in}}\text{-almost surely} \tag{17}$$

*where* $f_0(t)$ *solves* (15) *and satisfies*

$$\lim_{t \to \infty} f_0(t) = f \quad a.e. \ in \ \Omega. \tag{18}$$

*In addition, the limits in $n$ and $t$ commute, i.e. we also have* $\lim_{n \to \infty} \lim_{t \to \infty} f_n(t) = f$.

A detailed derivation of the LLN for gradient descent can be found in Sec. S1.2. The LLN should be understood as a guarantee that gradient descent reaches the optimal representation for initial conditions sampled iid from a smooth distribution with full support on $D \times \mathbb{R}$.

## 3.2 Central Limit Theorem and asymptotic fluctuations and error

To study the fluctuations around the optimal representation we look at the discrepancy between $f_n(t, \boldsymbol{x})$ and $f_0(t, \boldsymbol{x})$. These fluctuations are on the scale $O(n^{-1/2})$ initially and diminish as the optimization progresses to reach scale $O(n^{-1})$ or below, as summarized in the next two propositions.

**Proposition 3.2 (CLT for GD)** *Let $f_n(t)$ be as in Proposition 3.1. Then for any $t < \infty$ as $n \to \infty$, we have*

$$\lim_{n \to \infty} n^{-1/2} \left( f_n(t) - f_0(t) \right) = f_{1/2}(t) \qquad \text{in distribution} \tag{19}$$

*where $f_0(t)$ solves (15) and $f_{1/2}(t)$ is a Gaussian process with mean zero and some given covariance that satisfies $f_{1/2}(t) \to 0$ almost surely as $t \to \infty$.*

This result is derived in Sec. S2, where the covariance of $f_{1/2}(t)$ is also given (S46). Since $f_{1/2}(t)$ converges to zero as $t \to \infty$, it is useful to quantify the scale at which the fluctuations settle on long time scales:

**Proposition 3.3 (Asymptotic error for GD)** *Under the same conditions as those in Proposition 3.2, on any sequence $a_n > 0$ such that $a_n / \log n \to \infty$ as $n \to \infty$, we have*

$$\lim_{n \to \infty} n^{-\xi} \left( f_n(a_n) - f \right) = 0 \qquad \text{almost surely for any } \xi < 1 \tag{20}$$

This proposition characterizes the asymptotic error of the neural network, showing that it goes as $f_n = f + C n^{-1}$ for some constant $C \geq 0$. This scaling is more favorable than might be expected from the initial condition because the order of the error "heals" from $1/2$ to $1$ in the long time limit. That is, the error from the initial, non-optimal parameter selection decays during the optimization dynamics, becoming much more favorable at late times.

# 4 Stochastic gradient descent

We cannot typically evaluate the integrals required to compute $F(\boldsymbol{y})$ and $K(\boldsymbol{y}, \boldsymbol{y}')$. Instead, at each time step we estimate these functions using a small set of sample points $\{\boldsymbol{x}_i\}_{i=1}^P$ which we refer to as a batch of size $P$. Consequently, we introduce noise by sampling random data to make imperfect estimates of the gradient of the objective function. To estimate the gradient of the loss we use an unbiased estimator which is simply the sample mean over a collection or "batch" of $P$ points

$$E_P(\boldsymbol{z}) = \frac{n}{2P} \sum_{i=1}^P |f_n(\boldsymbol{x}_i, \boldsymbol{z}) - f(\boldsymbol{x}_i)|^2 \tag{21}$$

where, for simplicity, we write the parameters as a single vector $\boldsymbol{z} = (c_1, \boldsymbol{y}_1, \dots c_n, \boldsymbol{y}_n) \in (D \times \mathbb{R})^n$. Note that we have scaled the loss function by $n$ so that $\nabla E_P$ is $O(1)$ because our function representation is scaled by $n^{-1}$. The evolution equation of the corresponding dynamical variable $\boldsymbol{Z}(t)$ is

$$\boldsymbol{Z}(t + \Delta t) = \boldsymbol{Z}(t) - \Delta t \nabla E_P(\boldsymbol{Z}(t)). \tag{22}$$

The dynamics can be analyzed as a stochastic differential equation with a multiplicative noise term arising from the approximate evaluation of the gradient of the loss function. To derive this dynamical equation, we first need the covariance which we can write explicitly:

$$n^2 \int_\Omega (f_n - f)^2 \, \nabla f_n \otimes \nabla f_n d\mu - n^2 \nabla \ell(f, f_n) \otimes \nabla \ell(f, f_n) \equiv \frac{1}{P} R(\boldsymbol{z}). \tag{23}$$

where $f_n = f_n(\boldsymbol{x}, \boldsymbol{z})$ and $f = f(\boldsymbol{x})$. The discretized dynamics (22) is statistically equivalent to the stochastic differential equation

$$d\boldsymbol{Z} = -\nabla_{\boldsymbol{z}} E(\boldsymbol{Z}) dt + \sqrt{\theta} d\boldsymbol{B}(t, \boldsymbol{Z}) \tag{24}$$

where $E(\boldsymbol{z})$ is the energy (4) based on the exact loss, $\theta = \Delta t / P$, and the quadratic variation of the noise is $\langle d\boldsymbol{B}(t, \boldsymbol{z}), d\boldsymbol{B}(t, \boldsymbol{z}) \rangle = R(\boldsymbol{z}) dt$. The SDE (24) is *not* Langevin dynamics in the classical sense because the noise has spatiotemporal correlations. In our case, because new data is sampled at

every time step, there are no temporal correlations, which are a consequence of revisiting samples in a training set. Written in terms of $F$ and $K$, the parameters satisfy a collection of coupled SDEs that we can use to study the evolution of $\rho_n$,

$$
\begin{cases}
d\boldsymbol{Y}_i = C_i(t)\nabla F(\boldsymbol{Y}_i(t))\Delta t - \dfrac{1}{n}\sum_{j=1}^{n} C_i(t)C_j(t)\nabla K(\boldsymbol{Y}_i(t),\boldsymbol{Y}_j(t))\Delta t + d\boldsymbol{B}_i, \\[3mm]
dC_i = F(\boldsymbol{Y}_i(t))\Delta t - \dfrac{1}{n}\sum_{j=1}^{n} C_j(t)K(\boldsymbol{Y}_i(t),\boldsymbol{Y}_j(t))\Delta t + dB_i'
\end{cases}
\tag{25}
$$

where $\Delta t > 0$ is the time step. The time evolution of the parameter distribution can be derived by using the Itô formula, which in turn gives rise to a stochastic partial differential equation for the time-evolution of $\rho_n(t, c, \boldsymbol{y})$. This SPDE is

$$
\begin{aligned}
\partial_t \rho_n = \nabla \cdot (c\nabla U([\rho_n], \boldsymbol{y})\rho_n) + \partial_c \left( U([\rho_n], \boldsymbol{y})\rho_n \right) \\
+ \theta \mathcal{D}[\rho_n, \boldsymbol{y}, \boldsymbol{y}] + \sqrt{\theta}\left( \boldsymbol{\eta}(t, \boldsymbol{y}) + \eta(t, c) \right)
\end{aligned},
\tag{26}
$$

where $\mathcal{D}$ is a diffusive term given explicitly in Sec. S4.1 and which we do not reproduce here because it does not contribute in the subsequent scaling. This equation can be viewed as an extension of Dean's equation [20] to a setting with multiplicative noise. The noise terms $\boldsymbol{\eta}$ and $\eta$ (defined in Eq. S69) have a quadratic variation that diminishes as $f_n$ becomes close to $f$.

## 4.1 Law of large numbers

At first, it may appear that we could choose an arbitrary expansion in powers of $n^{-\alpha}$ for some $\alpha > 0$. However, as explained in Sec. S5, the expansion of $\rho_n \rho_n'$ contains terms of order $n^{-1}$, which constrains the choice of $\alpha$. To perform an expansion, we take $\theta \propto n^{-2\alpha}$ so that, in the limit $n \to \infty$, $\rho_0$ satisfies the same deterministic equation as in the case of gradient descent. This means that an analogous statement to Proposition 3.1 holds:

**Proposition 4.1 (LLN for SGD)** *Let* $f_n(t) = f_n(t, \boldsymbol{x}) = \sum_{i=1}^{n} C_i(t)\varphi(\boldsymbol{x}, \boldsymbol{Y}_i(t))$ *with* $\{\boldsymbol{Y}_i(t), C_i(t)\}_{i=1}^{n}$ *solution to (24) with* $\theta = an^{-2\alpha}$, $a > 0$ $\alpha \in (0, 1]$ *and initial condition where each pair* $(\boldsymbol{Y}_i(0), C_i(0))$ *is sampled independently from* $\rho_{\mathrm{in}} > 0$. *Then*

$$
\lim_{n \to \infty} f_n(t) = f_0(t)
\tag{27}
$$

*almost surely, where* $f_0(t)$ *solves (15). Furthermore,*

$$
\lim_{t \to \infty} f_0(t) = f \quad a.e. \ in \ \Omega.
\tag{28}
$$

*In addition the limits commute, i.e.* $\lim_{n \to \infty} \lim_{t \to \infty} f_n(t) = f$.

The Law of Large Numbers implies the universal approximation theorem, but notable additional information has emerged from our analysis. First, we emphasize that here we have obtained the representation as the limit of a stochastic gradient descent optimization procedure. Secondly, the PDE describing the time evolution of $f_0$ is independent of $n$, meaning the rate of convergence in time of $f_n$ does not depend on the number of parameters to leading order.

## 4.2 Asymptotic fluctuations and error

A remarkable feature of stochastic gradient descent is that the scale of fluctuations is controlled by the accuracy of the representation. Roughly, the closer $f_n$ is to $f$, the smaller the discrepancy in their gradients meaning that the variance of the noise term is also small. We make use of this property to assess the asymptotic error for stochastic gradient descent:

**Proposition 4.2 (Asymptotic error for SGD)** *Let* $f_n(t) = f_n(t, \boldsymbol{x})$ *be as in Proposition 4.1. Then for any* $a_n > 0$ *such that* $a_n / \log n \to \infty$ *as* $n \to \infty$, *we have*

$$
\lim_{n \to \infty} n^\alpha \left( f_n(a_n) - f \right) = 0 \qquad almost \ surely.
\tag{29}
$$

The discrepancy converges to zero almost surely with respect to the initial data as well as the statistics of the noise terms in (24). In terms of the loss function, we have

$$\ell(f, f_n(a_n)) = \tfrac{1}{2}\|f - f_0(a_n)\|^2 - n^{-\alpha} \langle f - f_0(a_n), f_\alpha(a_n)\rangle + \tfrac{1}{2}n^{-2\alpha}\|f_\alpha(a_n)\|^2 + o(n^{-\alpha}) \quad (30)$$

so that the following proposition holds:

**Proposition 4.3** *Under the same conditions as those in Proposition 4.2, the loss function satisfies*

$$\lim_{n\to\infty} n^\alpha \ell(f, f_n(a_n)) = 0 \qquad almost\ surely. \quad (31)$$

This means that the error at order $n^{-1}$ can be quenched by increasing the batch size or decreasing the time step as a function of the optimization time, e.g., setting $\alpha = 1$ by taking a batch of size $n^2$.

## 5   Numerical experiments

To test our results, we will use a function known for its complex features in high-dimensions: the spherical 3-spin model, which is a map from the $d - 1$ sphere of radius $\sqrt{d}$ to the reals $f :$ $S^{d-1}(\sqrt{d}) \to \mathbb{R}$, given by

$$f(\boldsymbol{x}) = \frac{1}{d} \sum_{p,q,r=1}^{d} a_{p,q,r} x_p x_q x_r, \qquad \boldsymbol{x} \in S^{d-1}(\sqrt{d}) \subset \mathbb{R}^d \quad (32)$$

where the coefficients $\{a_{p,q,r}\}_{p,q,r=1}^{d}$ are independent Gaussian random variables with mean zero and variance one. The function (32) is known to have a number of critical points that grows exponentially with the dimensionality $d$ [27, 6, 28]. We note that previous works have sought to draw a parallel between the glassy 3-spin function and generic loss functions [7], but we are not exploring such an analogy here. Rather, we simply use the function (32) as a difficult target for approximation by neural networks. That is, throughout this section, we train networks to learn $f$ with a particular realization of $a_{p,q,r}$ and study the accuracy of that representation as a function of the number of particles $n$. In Fig. 1 we show the representation error by computing the loss as well as the discrepancy between the target function and the neural network representation averaged over points at which the function is positive (or negative), i.e., $1/P \sum_{i=1}^{P} (f_n(\boldsymbol{x}_i) - f(\boldsymbol{x}_i)) \Theta(f(\boldsymbol{x}_i))$ where $\Theta$ is the Heaviside function.

**Single layer sigmoid / ReLU neural network** We consider the case that the nonlinear function $h(x)$ is $\max(0, x)$, the restricted linear unit or ReLU activation function frequently used in large scale applications of machine learning. In these experiments, we test the scaling in $d = 50$, prohibitively high dimensional for any grid based method. We trained the networks with batch size $P = 50$ using stochastic gradient descent with $n = i \times 10^4$ for $i = 1, \ldots, 6$. For the two smallest networks, we ran for $2 \times 10^6$ time steps with $\Delta t = 10^{-3}$ and then quenched with $P = 2500$ for $2 \times 10^5$ steps. For the largest networks, we used $\Delta t = 5 \times 10^{-4}$ to ensure stability and therefore doubled the number of steps so that the total training time remained fixed. Scaling data for the loss and the signed discrepancy are shown in Fig. 1. We also looked at sigmoid nonlinearities in $d = 10, 25$. These networks were trained as above but with $P = \lfloor n/5 \rfloor$ with a quench of $P^2$.

## 6   Conclusions and outlook

We have introduced a perspective based on particle distribution functions that enables asymptotic analysis of the optimization dynamics of neural networks. We have focused on the limit where the number of parameters $n \to \infty$, in which the objective function becomes convex and a stochastic partial differential equation describes the time evolution of the parameters. Our results emphasize that the optimal parameters in this limit are accessible via stochastic gradient descent (Proposition 4.1) and that fluctuations around the optimum can be controlled by modulating the batch size (Proposition 4.2). Surprisingly, the dynamical evolution does not depend on $n$, suggesting that the rate of convergence should be asymptotically independent of the number of parameters.

Our results do not address many features of neural network parameterization that merit further study exploiting the mathematical tools that have been developed for particle systems. In particular, the statements we have derived are insensitive to the details of network architecture, which is among the

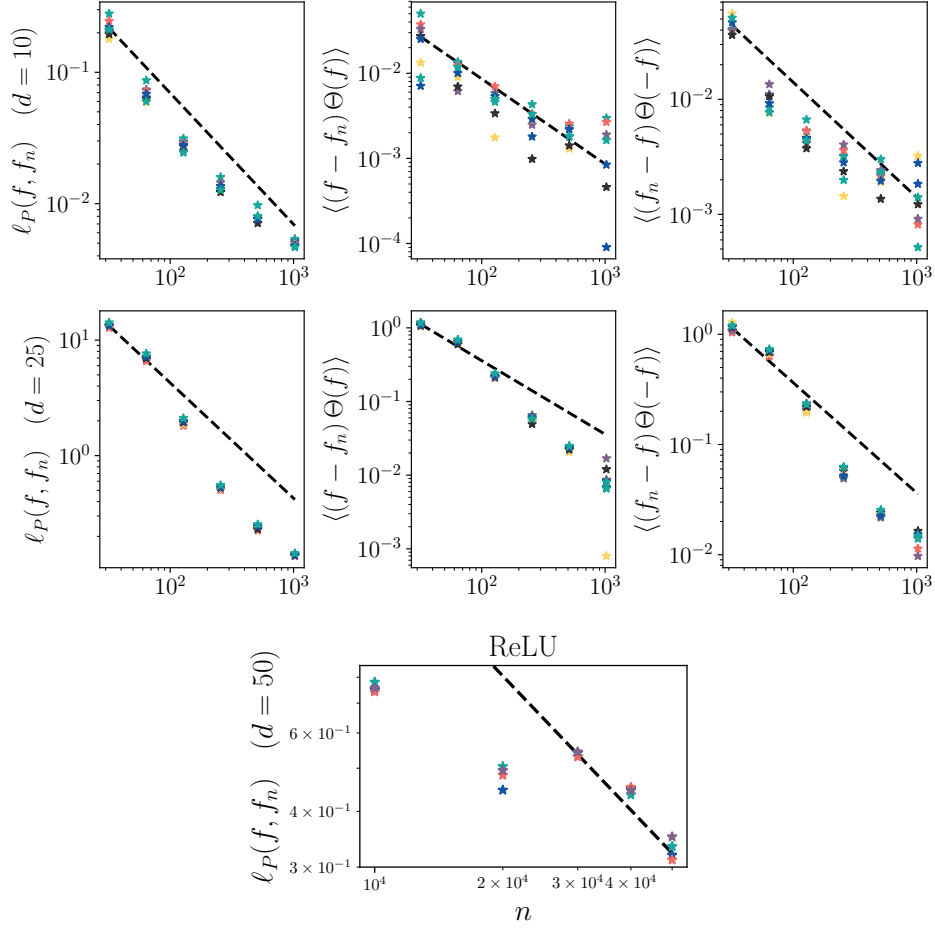

Figure 1: Large ReLU networks in high dimension ($d = 50$), and sigmoid neural networks in intermediate dimensions (bottom two rows). In all cases, we see linear scaling of the empirical loss averaged with $P = 10^6$. For the sigmoid neural networks, we also plot a measure of the discrepancy between the functions, which also scales as $O(n^{-1})$. In each plot, the error scaling as a function of the width of the network is plotted for 10 distinct random realizations of the function defined in (32) with different colored stars for each realization.

most important considerations when designing or using a neural network. It would also be beneficial to explore the ways in which regularizing processes, drop-out, for example, affect the convergence of the PDE. Developing a rigorous understanding of which kernels and which architectures are optimal for different types of target functions remains a compelling goal that appears within reach using the tools outlined here.

## Acknowledgments

We would like to thank Andrea Montanari and Matthieu Wyart for useful discussions regarding the fixed points of gradient flows in the Wasserstein metric. GMR was supported by the James S. McDonnell Foundation. EVE was supported by National Science Foundation (NSF) Materials Research Science and Engineering Center Program Award DMR-1420073; and by NSF Award DMS-1522767.

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
