[Supplementary Material · neurips_2018_SM.pdf]

# Supplementary Material: Parameters as interacting particles: long time convergence and asymptotic error scaling of neural networks

**Grant M. Rotskoff**
Courant Institute of Mathematical Sciences
New York University
rotskoff@cims.nyu.edu

**Eric Vanden-Eijnden**
Courant Institute of Mathematical Sciences
New York University
eve2@cims.nyu.edu

Here we give some justification to the results given in main text: for more details, we refer the reader to [1].

## 1  Law of large numbers for Gradient Descent

In most applications, the integrals defining $F$ and $K$ are intractable. However, it is useful to study the scaling of the representation error as $n \to \infty$ and $t \to \infty$ in the case that the parameters $\{\boldsymbol{Y}_i(t), C_i(t)\}_{i=1}^n$ evolve on the potential determined by the exact gradient of the loss function. In this case,

$$
\begin{cases}
\dot{\boldsymbol{Y}}_i = C_i \nabla F(\boldsymbol{Y}_i) - \dfrac{1}{n} \sum_{j=1}^n C_i C_j \nabla K(\boldsymbol{Y}_i, \boldsymbol{Y}_j), \\[2ex]
\dot{C}_i = F(\boldsymbol{Y}_i) - \dfrac{1}{n} \sum_{j=1}^n C_j K(\boldsymbol{Y}_i, \boldsymbol{Y}_j)
\end{cases}
\tag{1}
$$

for $i = 1, \ldots, n$. This dynamics can be expressed as a gradient flow on the energy function

$$
E(\boldsymbol{y}_1, c_1 \ldots, \boldsymbol{y}_n, c_n) = nC_f - \sum_{i=1}^n c_i F(\boldsymbol{y}_i) + \frac{1}{2n} \sum_{i,j=1}^n c_i c_j K(\boldsymbol{y}_i, \boldsymbol{y}_j),
\tag{2}
$$

which is precisely the loss function scaled by $n$. Due to the absence of noise in the dynamics, the initial conditions are a subtle consideration. We take each pair $\{\boldsymbol{Y}_i(0), C_i(0)\}_{i=1}^n$ independently from a density $\rho_{\mathrm{in}}(\boldsymbol{y}, c)$ that has full support on $D \times \mathbb{R}$ and is smooth in both its arguments. As shown in Ref. [2], then we have that $\rho_{\mathrm{in}} > 0$ in $D \times \mathbb{R}$ and $\int_{\mathbb{R}} c\rho_{\mathrm{in}}(\cdot, c)dc \in L^2(D)$. We denote the measure for the infinite set $\{(\boldsymbol{Y}_i(0), C_i(0))\}_{i \in \mathbb{N}}$ by $\mathbb{P}_{\mathrm{in}}$. In order to guarantee existence and uniqueness of the global solution to (1) we also make

**Assumption 1.1** *The kernel $\varphi(\cdot, \boldsymbol{x})$ is a continuously differentiable function of $\boldsymbol{y}$ for all $\boldsymbol{x} \in \Omega$.*

This assumption guarantees that the functions $F$ and $K$ are continuously differentiable in their arguments, and that the energy $E$ is continuously differentiable and coercive, i.e. for every $C \in \mathbb{R}$ the sub-level set

$$
E_C = \{(\boldsymbol{y}_1, c_1, \ldots, \boldsymbol{y}_n, c_n) \in (D \times \mathbb{R})^n : E(\boldsymbol{y}_1, c_1, \ldots, \boldsymbol{y}_n, c_n) \leq C\} \quad \text{is bounded.}
\tag{3}
$$

As a result, the solutions to the GD equations (1) exit and are unique for all times $t > 0$.

## 1.1 Empirical distribution and derivation of the nonlinear Liouville equation

In the main text, we refer to the nonlinear Liouville equation satisfied by the parameter density. The density is the large $n$ limit of the empirical distribution

$$\rho_n(t, \boldsymbol{y}, c) = \frac{1}{n} \sum_{i=1}^{n} \delta(c - C_i(t)) \delta(\boldsymbol{y} - \boldsymbol{Y}_i(t)). \tag{4}$$

We write the neural network representation as

$$f_n(t, \boldsymbol{x}) = \frac{1}{n} \sum_{i=1}^{n} C_i(t) \varphi(\boldsymbol{x}, \boldsymbol{Y}_i(t)) = \int_{D \times R} c \varphi(\boldsymbol{x}, \boldsymbol{y}) \rho_n(t, \boldsymbol{y}, c) d\boldsymbol{y} dc. \tag{5}$$

To derive the time evolution of $\rho_n$, we simply take the time derivative of (4),

$$\begin{aligned}
\partial_t \rho_n(t, \boldsymbol{y}, c) = &-\frac{1}{n} \sum_{i=1}^{n} \delta(c - C_i) \nabla \delta(\boldsymbol{y} - \boldsymbol{Y}_i) \cdot \dot{\boldsymbol{Y}}_i \\
&-\frac{1}{n} \sum_{i=1}^{n} \partial_c \delta(c - C_i) \delta(\boldsymbol{y} - \boldsymbol{Y}_i) \dot{C}_i
\end{aligned} \tag{6}$$

Pulling the derivatives in front of the sums and using (1) we can write this equation as

$$\begin{aligned}
\partial_t \rho_n(t, \boldsymbol{y}, c) = &-\nabla \cdot \left( \frac{1}{n} \sum_{i=1}^{n} \delta(c - C_i) \delta(\boldsymbol{y} - \boldsymbol{Y}_i(t)) \left( c \nabla F(\boldsymbol{y}) - \frac{1}{n} \sum_{j=1}^{n} c C_j \nabla K(\boldsymbol{y}, \boldsymbol{Y}_j) \right) \right) \\
&-\partial_c \left( \frac{1}{n} \sum_{i=1}^{n} \delta(c - C_i) \delta(\boldsymbol{y} - \boldsymbol{Y}_i) \left( F(\boldsymbol{y}) - \frac{1}{n} \sum_{j=1}^{n} C_j K(\boldsymbol{y}, \boldsymbol{Y}_j) \right) \right)
\end{aligned} \tag{7}$$

where we used the Dirac delta to replace $\boldsymbol{Y}_i$ by $\boldsymbol{y}$ and $C_i$ by $c$. We can now use the definition of $\rho_n$ to replace $\frac{1}{n} \sum_{i=1}^{n} \delta(c - C_i(t)) \delta(\boldsymbol{y} - \boldsymbol{Y}_i(t))$ by $\rho_n(t, \boldsymbol{y}, c)$. Using the fact that

$$\begin{aligned}
\frac{1}{n} \sum_{j=1}^{n} c C_j \nabla K(\boldsymbol{y}, \boldsymbol{Y}_j) &= \frac{1}{n} \sum_{j=1}^{n} \int_{D \times \mathbb{R}} c c' \nabla K(\boldsymbol{y}, \boldsymbol{y}') \delta(c' - C_j) \delta(\boldsymbol{y}' - \boldsymbol{Y}_j) d\boldsymbol{y}' dc' \\
&= \int_{D \times \mathbb{R}} c c' \nabla K(\boldsymbol{y}, \boldsymbol{y}') \rho_n(\boldsymbol{y}', c') d\boldsymbol{y}' dc'
\end{aligned} \tag{8}$$

and a similar calculation for $\frac{1}{n} \sum_{j=1}^{n} C_j K(\boldsymbol{y}, \boldsymbol{Y}_j)$, we see that the empirical distribution (4) satisfies

$$\partial_t \rho_n = \nabla \cdot (c \nabla U([\rho_n], \boldsymbol{y}) \rho_n) + \partial_c (U([\rho_n], \boldsymbol{y}) \rho_n), \tag{9}$$

where

$$U([\rho], \boldsymbol{y}) = -F(\boldsymbol{y}) + \int_{D \times \mathbb{R}} c' K(\boldsymbol{y}, \boldsymbol{y}') \rho(\boldsymbol{y}', c') d\boldsymbol{y}' dc' \tag{10}$$

When there is a diffusive term added, the nonlinear Liouville equation (9) is referred to as McKean-Vlasov equation[3, 4]. When there is noise, it is often referred to as Dean's equation [5].

## 1.2 Limit behavior and fluctuations scaling

All limits in this section should be understood in the weak sense—ultimately, we are interested in $f_n(t)$, not $\rho_n(t)$, and $f_n(t)$ is obtained by testing $\rho_n(t)$ against $c\varphi(\cdot, \boldsymbol{y})$, as in (5).

### 1.2.1 Zeroth order term—mean field limit

If we formally take the limit as $n \to \infty$ of the limiting equation for the distribution, we deduce that $\rho_n(t) \rightharpoonup \rho_0(t)$ where the arrow denotes weak convergence and $\rho_0(t)$ satisfies

$$\partial_t \rho_0 = \nabla \cdot (c \nabla U([\rho_0], \boldsymbol{y}) \rho_0) + \partial_c (U([\rho_0], \boldsymbol{y}) \rho_0), \tag{11}$$

This equation differs from the nonlinear Liouville equation (11) only in that it initial condition is smooth, $\rho_0(t=0) = \rho_{\text{in}}$—we show below that the solution to (11) also remains smooth for all $t > 0$. Notice that (11) can be written as

$$\partial_t \rho_0 = \nabla \cdot \left( \rho_0 \nabla \frac{\delta \mathcal{E}_0}{\delta \rho_0} \right) + \partial_c \left( \rho_0 \partial_c \frac{\delta \mathcal{E}_0}{\delta \rho_0} \right) \tag{12}$$

where $\mathcal{E}_0[\rho_0]$ is given by:

$$\mathcal{E}_0[\rho_0] = C_f - \int_{D \times \mathbb{R}} cF \rho_0 d\boldsymbol{y} dc + \tfrac{1}{2} \int_{(D \times \mathbb{R})^2} cc' K(\boldsymbol{y}, \boldsymbol{y}') \rho_0 \rho_0' d\boldsymbol{y} dc d\boldsymbol{y}' dc'$$

$$= \tfrac{1}{2} \int_\Omega \left( f(\boldsymbol{x}) - \int_{D \times \mathbb{R}} c\varphi(\boldsymbol{x}, \boldsymbol{y}) \rho_0 d\boldsymbol{y} dc \right)^2 d\mu(\boldsymbol{x}) \geq 0 \tag{13}$$

where we use the shorthands $\rho_0 = \rho_0(t, \boldsymbol{y}, c)$, $\rho_0' = \rho_0(t, \boldsymbol{y}', c')$, and similarly below. Note that the energy functional is quadratic in $\rho_0$ and hence is convex. The energy in (13) is the continuous limit of (2) scaled by $n^{-1}$, and (12) is the gradient decent flow on this energy in the Wasserstein metric. The energy can be expressed in terms of a signed density

$$G_0(\boldsymbol{y}) = \int_\mathbb{R} c\rho_0(\boldsymbol{y}, c) dc. \tag{14}$$

Indeed, (13) can be written as a functional of $G_0$ alone:

$$\mathcal{E}_0[\rho_0] = \hat{\mathcal{E}}_0 \left[ \int_\mathbb{R} c\rho_0(\cdot, c) dc \right] \quad \text{with} \quad \hat{\mathcal{E}}_0[G_0] = C_f - \int_D F G_0 d\boldsymbol{y} + \tfrac{1}{2} \int_{D^2} K(\boldsymbol{y}, \boldsymbol{y}') G_0 G_0' d\boldsymbol{y} d\boldsymbol{y}'. \tag{15}$$

The solution to the optimization problem is unique at the level of this signed density. To analyze the time evolution of the limiting density $\rho_0$, we study the time evolution of the neural network representation itself,

$$f_0(t, \boldsymbol{x}) = \int_{D \times \mathbb{R}} c\varphi(\boldsymbol{x}, \boldsymbol{y}) \rho_0(t, \boldsymbol{y}, c) d\boldsymbol{y} dc \tag{16}$$

Writing (11) as

$$\partial_t \rho_0 = \nabla \cdot \left( c \int_\Omega \nabla_{\boldsymbol{y}} \varphi(\boldsymbol{x}, \boldsymbol{y}) (f_0(t, \boldsymbol{x}) - f(\boldsymbol{x})) d\mu(\boldsymbol{x}) \rho_0 \right)$$

$$+ \partial_c \left( \int_\Omega \varphi(\boldsymbol{x}, \boldsymbol{y}) (f_0(t, \boldsymbol{x}) - f(\boldsymbol{x})) d\mu(\boldsymbol{x}) \rho_0 \right) \tag{17}$$

we deduce, using (16),

$$\partial_t f_0(t, \boldsymbol{x}) = \int_{D \times \mathbb{R}} c\varphi(\boldsymbol{x}, \boldsymbol{y}) \partial_t \rho_0(t, \boldsymbol{y}, c) d\boldsymbol{y} dc$$

$$= \int_{D \times \mathbb{R}} c\varphi(\boldsymbol{x}, \boldsymbol{y}) \nabla \cdot \left( c \int_\Omega \nabla_{\boldsymbol{y}} \varphi(\boldsymbol{x}, \boldsymbol{y}) (f_0(t, \boldsymbol{x}) - f(\boldsymbol{x})) d\mu(\boldsymbol{x}) \rho_0 \right) d\boldsymbol{y} dc \tag{18}$$

$$+ \int_{D \times \mathbb{R}} c\varphi(\boldsymbol{x}, \boldsymbol{y}) \partial_c \left( \int_\Omega \varphi(\boldsymbol{x}, \boldsymbol{y}) (f_0(t, \boldsymbol{x}) - f(\boldsymbol{x})) d\mu(\boldsymbol{x}) \rho_0 \right) d\boldsymbol{y} dc.$$

We define a symmetric kernel

$$M([\rho], \boldsymbol{x}, \boldsymbol{x}') = \int_{D \times \mathbb{R}} \left( c^2 \nabla_{\boldsymbol{y}} \varphi(\boldsymbol{x}, \boldsymbol{y}) \cdot \nabla_{\boldsymbol{y}} \varphi(\boldsymbol{x}', \boldsymbol{y}) + \varphi(\boldsymbol{x}, \boldsymbol{y}) \varphi(\boldsymbol{x}', \boldsymbol{y}) \right) \rho(\boldsymbol{y}, c) d\boldsymbol{y} dc. \tag{19}$$

And now, after integration by parts in $\boldsymbol{y}$ the first term and in $c$ the second, and interchanging the order of integration between $(\boldsymbol{y}, c)$ and $\boldsymbol{x}$ on both these terms, the equation for $\partial_t f_0$ can be written as

$$\partial_t f_0(t, \boldsymbol{x}) = - \int_\Omega M([\rho_0(t)], \boldsymbol{x}, \boldsymbol{x}') (f_0(t, \boldsymbol{x}') - f(\boldsymbol{x}')) d\mu(\boldsymbol{x}'). \tag{20}$$

We now observe that, given any $r \in L^2(\Omega, \mu)$ we have

$$\int_{\Omega^2} r(\boldsymbol{x}) r(\boldsymbol{x}') M([\rho], \boldsymbol{x}, \boldsymbol{x}') d\mu(\boldsymbol{x}) d\mu(\boldsymbol{x}') = \int_{D \times \mathbb{R}} \left( c^2 |\nabla R(\boldsymbol{y})|^2 + |R(\boldsymbol{y})|^2 \right) \rho(\boldsymbol{y}, c) d\boldsymbol{y} dc \tag{21}$$

where

$$R(\boldsymbol{y}) = \int_\Omega r(\boldsymbol{x})\varphi(\boldsymbol{x}, \boldsymbol{y})d\mu(\boldsymbol{x}). \tag{22}$$

As a result, (21) is non-negative so long as

$$\int_\mathbb{R} \rho(\cdot, c)dc > 0 \qquad \text{a.e. in } D, \tag{23}$$

and it can only be zero if $R = 0$ a.e. in $D$. By the assumptions we have placed on the kernel, this requires $r = 0$ a.e. in $\Omega$, which shows that $M([\rho], \boldsymbol{x}, \boldsymbol{x}')$ is positive-definite if (23) holds. It is easy to show that $\rho_0(t) > 0$ for all $t > 0$: Indeed we can turn (11) for the initial condition $\rho_0(0) = \rho_\text{in}$ into

$$\rho_0(t, \boldsymbol{y}, c) = \rho_\text{in}\left(\boldsymbol{Y}(-t), C(-t)\right) \exp\left(\int_0^t C(s - t)\Delta U([\rho_0(s)], \boldsymbol{Y}(s - t))ds\right) \tag{24}$$

where $(\boldsymbol{Y}(-t), C(-t))$ are the solution to the characteristic equations

$$\begin{cases} \dot{\boldsymbol{Y}}(t) = -C(t)\nabla U([\rho_0(t)], \boldsymbol{Y}(t)), & \boldsymbol{Y}(0) = \boldsymbol{y} \\ \dot{C}(t) = -U([\rho_0], \boldsymbol{Y}(t)), & C(0) = c \end{cases} \tag{25}$$

The representation formula (24) is readily verified by taking the time derivative of $\rho_0(t, \boldsymbol{Y}(t), C(t))$ and using the property $\boldsymbol{Y}(t, \boldsymbol{Y}(s), C(s)) = \boldsymbol{Y}(t + s)$ and $C(t, \boldsymbol{Y}(s), C(s)) = C(t + s)$ as well as (11) and (25). (24) is not explicit, since $U(t)$ depends on $\rho_0(t)$ through (23), but it shows that $\rho_0(t) > 0$ for all times $t > 0$ if $\rho_\text{in} > 0$. To show that this property also holds in the limit as $t \to \infty$, we can analyze the characteristic equations (25). These equations are gradient descent in the potential $cU(t, \boldsymbol{y}, [\rho_0(t)])$: because this potential is homogeneous of degree 1 in $c$, following [2] one can show that its flow cannot accumulate in any subset of $D \times \mathbb{R}$. As a result, $\rho_0(t) > 0$ for all $t > 0$ and in the limit as $t \to \infty$, which implies that $M([\rho_0(t), \boldsymbol{x}, \boldsymbol{x}')$ is always positive-definite, and the only stable fixed point of (20) is $f$. In other words, since $f_0(t)$ is the limit of $f_n(t)$ as $n \to \infty$, if we control the size of the fluctuations of $\rho_n(t)$ around $\rho_0(t)$ (which we do in Sec. 2.1), we have established:

**Proposition 1.2 (LLN)** *Let $f_n(t) = f_n(t, \boldsymbol{x})$ be given by (5) with $\{\boldsymbol{Y}_i(t), C_i(t)\}_{i=1}^n$ solution of (1) with initial condition drawn from $\mathbb{P}_\text{in}$. Then*

$$\lim_{n\to\infty} f_n(t) = f_0(t) \qquad \mathbb{P}_\text{in}\text{-almost surely} \tag{26}$$

*where $f_0(t)$ solves (20) and satisfies*

$$\lim_{t\to\infty} f_0(t) = f \quad \text{a.e. in } \Omega. \tag{27}$$

Proposition 1.2 indicates that the rate in time at which $\rho_0(t)$ converges towards its fixed point and $f_0(t)$ towards $f$ is independent of $n$ to leading order, since $n$ does not enter (11) or (20). This also implies that the limits (26) and (27) commute, i.e. we also have $\lim_{n\to\infty}\lim_{t\to\infty} f_n(t) = f$.

Notice that (20) confirms that $f_0(t)$ evolves on a quadratic landscape, namely the loss function. Indeed this equation can be written as

$$\partial_t f_0(t, \boldsymbol{x}) = -\int_\Omega M([\rho_0(t)], \boldsymbol{x}, \boldsymbol{x}')\, D_{f_0(t,\boldsymbol{x}')}\ell(f, f_0(t))d\mu(\boldsymbol{x}') \tag{28}$$

where $D_{f(\boldsymbol{x})}$ denotes the gradient with respect to $f(\boldsymbol{x})$ in the $L^2(\Omega, \mu)$-norm, i.e. given a functional $\mathcal{F}[f]$,

$$\forall h : \Omega \to \mathbb{R} \quad : \quad \lim_{z\to 0}\frac{d}{dz}\mathcal{F}[f + zh] = \langle h, D_f\mathcal{F}[f]\rangle_{L^2(\Omega,\mu)} = \int_\Omega h(\boldsymbol{x})D_{f(\boldsymbol{x})}\mathcal{F}[f]d\mu(\boldsymbol{x}). \tag{29}$$

In particular, $D_{f(\boldsymbol{x})}$ reduces to $\delta/\delta f(\boldsymbol{x})$ if $d\mu(\boldsymbol{x}) = d\boldsymbol{x}$.)

## 2 Gradient descent: Central limit theorem

In this section we provide a derivation of the CLT stated in the main text. To do so, we consider the fluctuations of $\rho_n(t)$ around $\rho_0(t)$. The scale of these fluctuations changes with time and to account for this effect, we define $\tilde{\rho}_\xi(t)$ via:

$$\rho_n = \rho_0 + n^{-\xi(t)}\tilde{\rho}_{\xi(t)}, \tag{30}$$

where the exponent $\xi(t)$ depends on $t$ as specified below. Explicitly, (30) means:

$$\tilde{\rho}_{\xi(t)}(t, \boldsymbol{y}, c) = n^{\xi(t)-1} \sum_{i=1}^{n} \left( \delta(\boldsymbol{y} - \boldsymbol{Y}_i(t)) \delta(c - C_i(t)) - \rho_0(t, \boldsymbol{y}, c) \right) \tag{31}$$

By the Central Limit Theorem, choosing $\xi(0) = 1/2$ sets scale of fluctuations around the initial conditions. Indeed, for any $\chi : D \times \mathbb{R} \to \mathbb{R}$, by the CLT under $\mathbb{P}_{\text{in}}$

$$\int_{D \times \mathbb{R}} \chi(\boldsymbol{y}, c) \tilde{\rho}_{\xi(0)}(0, \boldsymbol{y}, c) d\boldsymbol{y} dc = n^{-1/2} \sum_{i=1}^{n} \tilde{\chi}(\boldsymbol{Y}_i(0), C_i(0)) \to N(0, C_\chi) \quad \text{in law as } n \to \infty \tag{32}$$

where $N(0, C_\chi)$ denotes a Gaussian random variable with mean zero and variance $C_\chi$, and we defined

$$\tilde{\chi}(\boldsymbol{y}, c) = \chi(\boldsymbol{y}, c) - \int_{D \times \mathbb{R}} \chi(\boldsymbol{y}, c) \rho_{\text{in}}(\boldsymbol{y}, c) d\boldsymbol{y} dc, \qquad C_\chi = \int_{D \times \mathbb{R}} |\tilde{\chi}(\boldsymbol{y}, c)|^2 \, \rho_{\text{in}}(\boldsymbol{y}, c) d\boldsymbol{y} dc. \tag{33}$$

We can write (32) distributionally as

$$\tilde{\rho}_{\xi(0)}(0, \boldsymbol{y}, c) \rightharpoonup N\left(0, \rho_{\text{in}}(\boldsymbol{y}, c) \delta(\boldsymbol{y} - \boldsymbol{y}') \delta(c - c')\right) \quad \text{in law as } n \to \infty \tag{34}$$

We derive an equation for $\tilde{\rho}_{\xi(t)}$ by subtracting (11) from (9) and using (30):

$$\partial_t \tilde{\rho}_{\xi(t)} = \nabla \cdot \left( -c \nabla F \tilde{\rho}_{\xi(t)} + \int_{D \times \mathbb{R}} cc' \nabla K(\boldsymbol{y}, \boldsymbol{y}') \left( \tilde{\rho}'_{\xi(t)} \rho_0 + \rho'_0 \tilde{\rho}_{\xi(t)} + n^{-\xi(t)} \tilde{\rho}'_{\xi(t)} \tilde{\rho}_{\xi(t)} \right) d\boldsymbol{y}' dc' \right)$$

$$+ \partial_c \left( -F \tilde{\rho}_{\xi(t)} + \int_{D \times \mathbb{R}} c' K(\boldsymbol{y}, \boldsymbol{y}') \left( \tilde{\rho}'_{\xi(t)} \rho_0 + \rho'_0 \tilde{\rho}_{\xi(t)} + n^{-\xi(t)} \tilde{\rho}'_{\xi(t)} \tilde{\rho}_\xi \right) d\boldsymbol{y}' dc' \right)$$

$$+ \dot{\xi}(t) \log n \, \tilde{\rho}_{\xi(t)}. \tag{35}$$

In order to take the limit as $n \to \infty$ of this equation, we need to consider carefully the behavior of the the factors in (35) that contain $n$ explicitly, that is, $n^{-\xi(t)} \tilde{\rho}_{\xi(t)} \tilde{\rho}'_{\xi(t)}$ and $\dot{\xi}(t) \log n \, \tilde{\rho}_{\xi(t)}$. Consider the latter first. If we set

$$\dot{\xi}(t) \log n = o(1) \tag{36}$$

the last term at the right hand side of (35) is higher order. Note that (36) means that we can vary $\xi(t)$, but only slowly. For the factor $n^{-\xi(t)} \tilde{\rho}_{\xi(t)} \tilde{\rho}'_{\xi(t)}$, a direct calculation shows that, for any $p \in \mathbb{N}$ and $\xi \in \mathbb{R}$,

$$\mathbb{E}_{\text{in}} \left( n^{-\xi} \int_{(D \times \mathbb{R})^2} \chi(\boldsymbol{y}, c) \chi(\boldsymbol{y}', c') \tilde{\rho}_\xi \tilde{\rho}'_\xi d\boldsymbol{y} dc d\boldsymbol{y}' dc' \right)^p = O\left(n^{(\xi-1)p}\right) \tag{37}$$

where $\mathbb{E}_{\text{in}}$ denotes expectation with respect to $\mathbb{P}_{\text{in}}$—for example, if $p = 1$, this expectation is $n^{(\xi-1)} C_\chi$ where $C_\chi$ is given in (33). The expectation calculation implies that $\mathbb{P}_{\text{in}}$-almost surely as $n \to \infty$, $n^{-\xi} \tilde{\rho}_n(0) \tilde{\rho}'_n(0) \rightharpoonup 0$ at $t = 0$ if $\xi < 1$. We can now argue that this statement also holds at times $t > 0$ if (36) holds. To this end we write (35) compactly as

$$\partial_t \tilde{\rho}_{\xi(t)} = L \tilde{\rho}_{\xi(t)} + R_{\xi(t)} + \dot{\xi}(t) \log n \, \tilde{\rho}_{\xi(t)} \tag{38}$$

where $L \tilde{\rho}_n$ contains the terms at the right hand side of (35) that are linear in $\tilde{\rho}_{\xi(t)}$, and $R_{\xi(t)}$ contains the terms involving $n^{-\xi(t)} \tilde{\rho}_{\xi(t)} \tilde{\rho}'_{\xi(t)}$. In order to control the $R_{\xi(t)}$ term, we can write an equation for $n^{-\xi(t)} \tilde{\rho}_{\xi(t)} \tilde{\rho}'_{\xi(t)}$: this equation is of the form (38) with an additional linear term involving $L'$ (same as $L$ but acting on $(\boldsymbol{y}', c')$), the source term $R_{\xi(t)}$ replaced by one involving $n^{-2\xi(t)} \tilde{\rho}_{\xi(t)} \tilde{\rho}'_{\xi(t)} \tilde{\rho}''_{\xi(t)}$, and the last term in (38) replaced by $2\dot{\xi}(t) \log n \, \tilde{\rho}_{\xi(t)} \tilde{\rho}'_{\xi(t)}$: a calculation similar to the one that gives (37) indicates that at $t = 0$ this source term is higher order than the rest and goes to zero $\mathbb{P}_{\text{in}}$-almost surely as $n \to \infty$. The same is true for $2\dot{\xi}(t) \log n \, \tilde{\rho}_{\xi(t)} \tilde{\rho}'_{\xi(t)}$ if (36) holds. We can then derive equations for $n^{-2\xi(t)} \tilde{\rho}_{\xi(t)} \tilde{\rho}'_{\xi(t)} \tilde{\rho}''_{\xi(t)}$ and so on, and each time reach the same conclusion: they involve a linear part made of operators $L$, $L'$, etc. and a remainder that is higher order.

This argument implies that, as long as (36) holds and $\xi(t) < 1$ at all times, $\tilde{\rho}_{\xi(t)}$ has a limit as $n \to \infty$. If we take this limit at any fixed time, then (36) implies that $\xi(t) = \xi(0) = \frac{1}{2}$ as $n \to \infty$, that is, we

have remained on the original scale set by $\mathbb{P}_{\text{in}}$. On that scale, we have $\tilde{\rho}_{1/2}(t) \rightharpoonup \rho_{1/2}(t)$ as $n \to \infty$, where $\rho_{1/2}(t)$ solves

$$
\begin{aligned}
\partial_t \rho_{1/2} = \nabla \cdot \left( -c\nabla F \rho_{1/2} + \int_{D \times \mathbb{R}} cc' \nabla K(\boldsymbol{y}, \boldsymbol{y}') \left( \rho'_{1/2} \rho_0 + \rho'_0 \rho_{1/2} \right) d\boldsymbol{y}' dc' \right) \\
+ \partial_c \left( -F \rho_{1/2} + \int_{D \times \mathbb{R}} c' K(\boldsymbol{y}, \boldsymbol{y}') \left( \rho'_{1/2} \rho_0 + \rho'_0 \rho_{1/2} \right) d\boldsymbol{y}' dc' \right)
\end{aligned}
\tag{39}
$$

this equation should be solved with the Gaussian initial conditions read from (34):

$$
\rho_{1/2}(0, \boldsymbol{y}, c) = N\left(0, \rho_{\text{in}}(\boldsymbol{y}, c) \delta(\boldsymbol{y} - \boldsymbol{y}') \delta(c - c')\right).
\tag{40}
$$

Note that since the mean of $\rho_{1/2}$ is zero initially and (39) is linear, this mean remains zero for all times, and we can focus in the evolution of its covariance. Denoting this covariance by

$$
\omega_{1/2}(t, \boldsymbol{y}, c, \boldsymbol{y}', c') = \mathbb{E}_{\text{in}} \rho_{1/2}(t, \boldsymbol{y}, c) \rho_{1/2}(t, \boldsymbol{y}', c'),
\tag{41}
$$

from (39) it satisfies

$$
\begin{aligned}
\partial_t \omega_{1/2} = \nabla \cdot \left( -c\nabla F \omega_{1/2} + \int_{D \times \mathbb{R}} cc'' \nabla K(\boldsymbol{y}, \boldsymbol{y}'') \left( \omega_{1/2} \rho_0 + \omega_{1/2} \rho_0'' \right) d\boldsymbol{y}'' dc'' \right) \\
+ \partial_c \left( -F \omega_{1/2} + \int_{D \times \mathbb{R}} c'' K(\boldsymbol{y}, \boldsymbol{y}'') \left( \omega_{1/2} \rho_0 + \omega_{1/2} \rho_0'' \right) d\boldsymbol{y}'' dc'' \right) \\
+ \nabla' \cdot \left( -c'\nabla' F' \omega_{1/2} + \int_{D \times \mathbb{R}} c'c'' \nabla K(\boldsymbol{y}, \boldsymbol{y}'') \left( \omega_{1/2} \rho_0' + \omega_{1/2} \rho_0'' \right) d\boldsymbol{y}'' dc'' \right) \\
+ \partial_{c'} \left( -F' \omega_{1/2} + \int_{D \times \mathbb{R}} c'' K(\boldsymbol{y}', \boldsymbol{y}'') \left( \omega_{1/2} \rho_0' + \omega_{1/2} \rho_0'' \right) d\boldsymbol{y}'' dc'' \right)
\end{aligned}
\tag{42}
$$

where we use the shorthand $\omega_{1/2}\rho_0 = \omega_{1/2}(t, \boldsymbol{y}', c', \boldsymbol{y}'', c'')\rho_0(t, \boldsymbol{y}, c)$, $\omega_{1/2}\rho_0' = \omega_{1/2}(t, \boldsymbol{y}'', c'', \boldsymbol{y}, c)\rho_0(t, \boldsymbol{y}', c')$, and $\omega_{1/2}\rho_0'' = \omega_{1/2}(t, \boldsymbol{y}, c, \boldsymbol{y}', c')\rho_0(t, \boldsymbol{y}'', c'')$. The initial condition for (42) is

$$
\omega_{1/2}(0, \boldsymbol{y}, c, \boldsymbol{y}', c') = \rho_{\text{in}}(\boldsymbol{y}, c) \delta(\boldsymbol{y} - \boldsymbol{y}') \delta(c - c').
\tag{43}
$$

The existence of the weak limit of $\tilde{\rho}_{1/2}$ even with $\xi(t) = \xi(0) = \frac{1}{2}$ is enough to confirm that $\rho_n(t) \rightharpoonup \rho_0(t)$ as $n \to \infty$, where $\rho_0(t)$ solves (11). It also gives the scaling of fluctuations around that limit at finite time: as $n \to \infty$, we have

$$
n^{1/2} \left( \rho_n(t, \boldsymbol{y}, c) - \rho_0(t, \boldsymbol{y}, c) \right) \rightharpoonup \rho_{1/2}(t, \boldsymbol{y}, c), \qquad \text{in law}
\tag{44}
$$

where $\rho_{1/2}(t)$ is the zero-mean Gaussian process whose covariance $\omega_{1/2}(t)$ solves (42). It is useful to formulate this Central Limit Theorem in terms of $f_n(t)$:

**Proposition 2.1 (CLT)** *Let $f_n(t) = f_n(t, \boldsymbol{x})$ be given by (5) with $\{Y_i(t), C_i(t)\}_{i=1}^n$ solution of (1) with initial condition drawn from $\mathbb{P}_{\text{in}}$. Then as $n \to \infty$:*

$$
n^{1/2} \left( f_n(t) - f_0(t) \right) \to f_{1/2}(t), \qquad \text{in law}
\tag{45}
$$

*where $f_{1/2}(t)$ is the zero-mean Gaussian process whose covariance is given by*

$$
\mathbb{E}_{\text{in}} f_{1/2}(t, \boldsymbol{x}) f_{1/2}(t, \boldsymbol{x}') = \int_{(D \times \mathbb{R})^2} cc' \varphi(\boldsymbol{x}, \boldsymbol{y}) \varphi(\boldsymbol{x}, \boldsymbol{y}) \omega_{1/2}(t, \boldsymbol{y}, c, \boldsymbol{y}', c') d\boldsymbol{y} dc d\boldsymbol{y}' dc'
\tag{46}
$$

## 3 Gradient descent: asymptotic error

It is useful to consider different values of $\xi(t)$ in (30) to get a better handle on the size of the fluctuations at long times. This is needed because, as shown below, the solution to (42) converges to zero as $t \to \infty$, i.e. the fluctuation disappear on this scale. To analyze on which scale these fluctuations settle as $t \to \infty$, we set

$$
\xi(t) = \bar{\xi}(t/a_n) \quad \text{with} \quad \lim_{n \to \infty} a_n / \log n = \infty, \quad \bar{\xi}(0) = \tfrac{1}{2}, \quad \bar{\xi}(s) < 1 \;\; \forall s > 0
\tag{47}
$$

so that $\dot{\xi}(t) = \bar{\xi}'(t/a_n)/a_n$ and satisfies (36). If we then set the time to be $a_n$, we can conclude that $\tilde{\rho}_{\xi(a_n)}(a_n) \rightharpoonup \rho_{\bar{\xi}}$, where $\bar{\xi} = \bar{\xi}(1)$ and $\rho_{\bar{\xi}}$ solves

$$
\begin{aligned}
0 = \nabla \cdot &\left( -c\nabla F \rho_{\bar{\xi}} + \int_{D\times\mathbb{R}} cc'\nabla K(\boldsymbol{y}, \boldsymbol{y}') \left( \rho'_{\bar{\xi}}\rho_0 + \rho'_0\rho_{\bar{\xi}} \right) d\boldsymbol{y}' dc' \right) \\
&+ \partial_c \left( -F\rho_{\bar{\xi}} + \int_{D\times\mathbb{R}} c' K(\boldsymbol{y}, \boldsymbol{y}') \left( \rho'_{\bar{\xi}}\rho_0 + \rho'_0\rho_{\bar{\xi}} \right) d\boldsymbol{y}' dc' \right)
\end{aligned}
\tag{48}
$$

where $\rho_0$ in this equation is understood as $\lim_{n\to\infty} \rho_0(a_n) = \lim_{t\to\infty} \rho_0(t)$, a limit for which we have already shown existence. Since (48) is linear and homogeneous in $\rho_{\bar{\xi}}$, either zero is the stable fixed point of this equation, and it means that the size of the fluctuations are bounded from above by $O(n^{-\bar{\xi}})$ asymptotically: $\tilde{\rho}_{\bar{\xi}}(a_n) \rightharpoonup 0$; or zero is an unstable fixed point of (48), and these fluctuations go to infinity even on the scale $O(n^{-1/2})$.

To see that the first scenario holds, write (48) as

$$
0 = \nabla \cdot \left( \int_{D\times\mathbb{R}} cc'\nabla K(\boldsymbol{y}, \boldsymbol{y}')\rho'_{\bar{\xi}}\rho_0 d\boldsymbol{y}' dc' \right) + \partial_c \left( \int_{D\times\mathbb{R}} c' K(\boldsymbol{y}, \boldsymbol{y}')\rho'_{\bar{\xi}}\rho_0 d\boldsymbol{y}' dc' \right)
\tag{49}
$$

Proceeding as we did to derive (20) we can write an equation for

$$
f_{\bar{\xi}}(\boldsymbol{x}) = \int_{D\times\mathbb{R}} c\varphi(\boldsymbol{x}, \boldsymbol{y})\rho_{\bar{\xi}}(\boldsymbol{y}, c)d\boldsymbol{y}dc
\tag{50}
$$

which is

$$
0 = -\int_{\Omega} M([\rho_0], \boldsymbol{x}, \boldsymbol{x}')f_{\bar{\xi}}(\boldsymbol{x}')d\mu(\boldsymbol{x}')
\tag{51}
$$

where $M([\rho_0], \boldsymbol{x}, \boldsymbol{x}')$ is the kernel defined in (19) evaluated on $\rho = \rho_0 = \lim_{t\to\infty} \rho_0(t)$. Since this kernel is positive-definite at all times $t > 0$ and in the limit as $t \to \infty$, and the only fixed point of (51) is zero so that

$$
f_{\bar{\xi}} = 0.
\tag{52}
$$

This also implies that $\rho_{\bar{\xi}} = 0$ is a stable fixed point of (49). Summarizing, we have established:

**Proposition 3.1 (Asymptotic error)** *Let $f_n(t) = f_n(t, \boldsymbol{x})$ be given by (5) with $\{\boldsymbol{Y}_i(t), C_i(t)\}_{i=1}^n$ solution of (1) with initial condition drawn from $\mathbb{P}_{in}$. Then for any $a_n > 0$ such that $a_n/\log n \to \infty$ as $n \to \infty$, we have*

$$
\lim_{n\to\infty} n^{\bar{\xi}} \left( f_n(a_n) - f \right) = 0 \qquad \text{almost surely for any } \bar{\xi} < 1
\tag{53}
$$

The fluctuations at scale $O(n^{-1/2})$ of $f_n(t)$ around $f_0(t)$ that were present initially decrease in amplitude as time progresses, and become $O(n^{-1})$ as $t \to \infty$, a type of self-healing.

## 4  Stochastic Gradient Descent: Law of large numbers

The SGD dynamics is

$$
\begin{cases}
\boldsymbol{Y}_i^P(t + \Delta t) = \boldsymbol{Y}_i^P(t) + C_i^P(t)\nabla F_P(t, \boldsymbol{Y}_i^P(t))\Delta t - \dfrac{1}{n}\displaystyle\sum_{j=1}^n C_i^P(t)C_j^P(t)\nabla K_P(t, \boldsymbol{Y}_i^P(t), \boldsymbol{Y}_j^P(t))\Delta t \\[3mm]
C_i^P(t + \Delta t) = C_i(t) + F_P(t, \boldsymbol{Y}_i^P(t))\Delta t - \dfrac{1}{n}\displaystyle\sum_{j=1}^n C_j^P(t)K_P(t, \boldsymbol{Y}_i^P(t), \boldsymbol{Y}_j^P(t))\Delta t
\end{cases}
\tag{54}
$$

where $\Delta t > 0$ is the time-step and

$$
F_P(t, \boldsymbol{y}) = \frac{1}{P}\sum_{p=1}^P f(\boldsymbol{X}_p(t))\varphi(\boldsymbol{X}_p(t), \boldsymbol{y}), \qquad K_P(t, \boldsymbol{y}, \boldsymbol{y}') = \frac{1}{P}\sum_{p=1}^P \varphi(\boldsymbol{X}_p(t), \boldsymbol{y})\varphi(\boldsymbol{X}_p(t), \boldsymbol{y}')
\tag{55}
$$

in which $\{\boldsymbol{X}_p(t)\}_{p=1}^P$ are $P$ iid variables which are redrawn from $\mu$ independently at every time step $t$.

## 4.1 Limiting stochastic differential equation (SDE)

First, we denote the set of all particles as

$$\boldsymbol{z} = (\boldsymbol{z}_1, \ldots, \boldsymbol{z}_n) = (\boldsymbol{y}_1, c_1, \ldots, \boldsymbol{y}_n, c_n) \in (D \times \mathbb{R})^n, \quad \boldsymbol{z}_i = (\boldsymbol{y}_i, c_i) \in D \times \mathbb{R} \quad i = 1, \ldots, n \tag{56}$$

where

$$f_n(\boldsymbol{z}) = \frac{1}{n} \sum_{i=1}^{n} c_i \varphi(\cdot, \boldsymbol{y}_i), \qquad f_n(\boldsymbol{x}, \boldsymbol{z}) = \frac{1}{n} \sum_{i=1}^{n} c_i \varphi(\boldsymbol{x}, \boldsymbol{y}_i). \tag{57}$$

Using this notation, (54) is just

$$\boldsymbol{Z}(t + \Delta t) = \boldsymbol{Z}(t) - \Delta t \nabla_{\boldsymbol{z}} E_P(\boldsymbol{Z}(t)) \tag{58}$$

where $E_P(\boldsymbol{z})$ is the approximation of the loss function obtained with a batch of $P$ independent samples $\{\boldsymbol{X}_p(t)\}_{p=1}^{P}$ drawn from $\mu$ and scaled by $n$:

$$E_P(\boldsymbol{z}) = \frac{n}{2P} \sum_{p=1}^{P} |f(\boldsymbol{X}_p) - f_n(\boldsymbol{X}_p, \boldsymbol{z})|^2 \tag{59}$$

The estimator is unbiased and $E_P(\boldsymbol{z})$ has expectation $n\ell(f, f_n(\boldsymbol{z}))$. Explicitly,

$$\nabla_{\boldsymbol{z}} E_P(\boldsymbol{z}) = \frac{n}{P} \sum_{p=1}^{P} (f_n(\boldsymbol{X}_p, \boldsymbol{z}) - f(\boldsymbol{X}_p)) \nabla_{\boldsymbol{z}} f_n(\boldsymbol{X}_p, \boldsymbol{z}). \tag{60}$$

To characterize the noise of the SDE, we need to compute the covariance

$$\mathbb{E} \left( \nabla_{\boldsymbol{z}} (E_P(\boldsymbol{z}) - n\ell(f, f_n(\boldsymbol{z}))) \right) \otimes \left( \nabla_{\boldsymbol{z}} (E_P(\boldsymbol{z}') - n\ell(f, f_n(\boldsymbol{z}'))) \right) = \frac{1}{P} R(\boldsymbol{z}) \tag{61}$$

with

$$R(\boldsymbol{z}) = n^2 \int_{\Omega} |f(\boldsymbol{x}) - f_n(\boldsymbol{x}, \boldsymbol{z})|^2 \nabla_{\boldsymbol{z}} f_n(\boldsymbol{x}, \boldsymbol{z}) \otimes \nabla_{\boldsymbol{z}} f_n(\boldsymbol{x}, \boldsymbol{z}) d\mu(\boldsymbol{x})$$
$$- n^2 \nabla_{\boldsymbol{z}} \ell(f, f_n(\boldsymbol{z})) \otimes \nabla_{\boldsymbol{z}} \ell(f, f_n(\boldsymbol{z})) \tag{62}$$

We can write this covariance in block tensor notation

$$R_{i,j}(\boldsymbol{z}) = \begin{pmatrix} c_i c_j A_2([f - f_n], \boldsymbol{y}_i, \boldsymbol{y}_j) & c_i A_1([f - f_n], \boldsymbol{y}_i, \boldsymbol{y}_j) \\ c_j A_1([f - f_n], \boldsymbol{y}_j, \boldsymbol{y}_i) & A_0([f - f_n], \boldsymbol{y}_i, \boldsymbol{y}_j) \end{pmatrix} \tag{63}$$

where

$$A_0([f], \boldsymbol{y}, \boldsymbol{y}') = \int_{\Omega} |f(\boldsymbol{x})|^2 \varphi(\boldsymbol{x}, \boldsymbol{y}) \varphi(\boldsymbol{x}, \boldsymbol{y}') d\mu(\boldsymbol{x})$$
$$- \int_{\Omega} f(\boldsymbol{x}) \varphi(\boldsymbol{x}, \boldsymbol{y}) d\mu(\boldsymbol{x}) \int_{\Omega} f(\boldsymbol{x}) \varphi(\boldsymbol{x}, \boldsymbol{y}') d\mu(\boldsymbol{x})$$

$$A_1([f], \boldsymbol{y}, \boldsymbol{y}') = \int_{\Omega} |f(\boldsymbol{x})|^2 \nabla_{\boldsymbol{y}} \varphi(\boldsymbol{x}, \boldsymbol{y}) \varphi(\boldsymbol{x}, \boldsymbol{y}') d\mu(\boldsymbol{x})$$
$$- \int_{\Omega} f(\boldsymbol{x}) \nabla_{\boldsymbol{y}} \varphi(\boldsymbol{x}, \boldsymbol{y}) d\mu(\boldsymbol{x}) \int_{\Omega} f(\boldsymbol{x}) \varphi(\boldsymbol{x}, \boldsymbol{y}') d\mu(\boldsymbol{x}) \tag{64}$$

$$A_2([f], \boldsymbol{y}, \boldsymbol{y}') = \int_{\Omega} |f(\boldsymbol{x})|^2 \nabla_{\boldsymbol{y}} \varphi(\boldsymbol{x}, \boldsymbol{y}) \otimes \nabla_{\boldsymbol{y}'} \varphi(\boldsymbol{x}, \boldsymbol{y}') d\mu(\boldsymbol{x})$$
$$- \int_{\Omega} f(\boldsymbol{x}) \nabla_{\boldsymbol{y}} \varphi(\boldsymbol{x}, \boldsymbol{y}) d\mu(\boldsymbol{x}) \otimes \int_{\Omega} f(\boldsymbol{x}) \nabla_{\boldsymbol{y}'} \varphi(\boldsymbol{x}, \boldsymbol{y}') d\mu(\boldsymbol{x})$$

where $A_0 \in \mathbb{R}$, $A_1 \in \mathbb{R}^n$, and $A_2 \in \mathbb{R}^n \times \mathbb{R}^n$.

The discrete SGD equation (54) resembles an Euler-Maruyama integration scheme, but the noise scales as $\Delta t$ rather than $\sqrt{\Delta t}$. The corresponding SDE, which is equivalent to SGD up to discretization errors, is

$$d\boldsymbol{Z} = n\nabla_{\boldsymbol{z}} \ell(f, f_n(\boldsymbol{Z}))dt + \sqrt{\theta} d\boldsymbol{B} \tag{65}$$

where $\theta = \Delta t/P$ and $d\boldsymbol{B}$ is a white-noise process with quadratic variation

$$\langle d\boldsymbol{B}, d\boldsymbol{B} \rangle = R(\boldsymbol{Z})dt. \tag{66}$$

In the parameter variables, (65) reads

$$
\begin{cases}
d\boldsymbol{Y}_i = C_i \nabla F(\boldsymbol{Y}_i)dt - \dfrac{1}{n}\sum_{j=1}^{n} C_i C_j \nabla K(\boldsymbol{Y}_i, \boldsymbol{Y}_j)dt + \sqrt{\theta}d\boldsymbol{B}_i, \\[4mm]
dC_i = F(\boldsymbol{Y}_i)dt - \dfrac{1}{n}\sum_{j=1}^{n} C_j K(\boldsymbol{Y}_i, \boldsymbol{Y}_j)dt + \sqrt{\theta}dB_i'
\end{cases} \tag{67}
$$

where $\{d\boldsymbol{B}_i, dB_i'\}_{i=1}^{n}$ are a white-noise processes with quadratic variation

$$
\begin{aligned}
\langle d\boldsymbol{B}_i, d\boldsymbol{B}_j \rangle &= C_i C_j A_2([f - f_n], \boldsymbol{Y}_i, \boldsymbol{Y}_j)dt, \\
\langle d\boldsymbol{B}_i, dB_j' \rangle &= C_i A_1([f - f_n], \boldsymbol{Y}_i, \boldsymbol{Y}_j)dt, \\
\langle dB_i', dB_j' \rangle &= A_0([f - f_n], \boldsymbol{Y}_i, \boldsymbol{Y}_j)dt.
\end{aligned} \tag{68}
$$

## 4.2 Dean's equation for particles with correlated noise

We analyze (67) instead of the discrete version (54), noting that the continuous limit is never achieved in practice due to the finite step sizes. Applying Itô's formula to (4) when $\{\boldsymbol{Y}_i(t), C_i(t)\}_{i=1}^{n}$ satisfy (67), we see

$$
\begin{aligned}
d\rho_n(t, \boldsymbol{y}, c) = &-\frac{1}{n}\sum_{i=1}^{n} \delta(c - C_i)\nabla\delta(\boldsymbol{y} - \boldsymbol{Y}_i)\cdot d\boldsymbol{Y}_i \\
&- \frac{1}{n}\sum_{i=1}^{n} \partial_c\delta(c - C_i)\delta(\boldsymbol{y} - \boldsymbol{Y}_i)dC_i \\
&+ \frac{\theta}{2n}\sum_{i=1}^{n} \delta(c - C_i)\nabla\nabla\delta(\boldsymbol{y} - \boldsymbol{Y}_i) : C_i C_i A_2([f - f_n], \boldsymbol{Y}_i, \boldsymbol{Y}_i)dt \\
&+ \frac{\theta}{n}\sum_{i=1}^{n} \partial_c^2\delta(c - C_i)\nabla\delta(\boldsymbol{y} - \boldsymbol{Y}_i)\cdot C_i A_1([f - f_n], \boldsymbol{Y}_i, \boldsymbol{Y}_i)dt \\
&+ \frac{\theta}{2n}\sum_{i=1}^{n} \partial_c^2\delta(c - C_i)\delta(\boldsymbol{y} - \boldsymbol{Y}_i)A_0([f - f_n], \boldsymbol{Y}_i, \boldsymbol{Y}_i)dt
\end{aligned} \tag{69}
$$

We use (4) to write $d\boldsymbol{Y}_i$ and $dC_i$, the drift terms that emerge can be treated as we did to derive (9). The noise term in (69) is

$$-\frac{1}{n}\sum_{i=1}^{n} \delta(c - C_i)\nabla\delta(\boldsymbol{y} - \boldsymbol{Y}_i)\cdot d\boldsymbol{B}_i - \frac{1}{n}\sum_{i=1}^{n} \partial_c\delta(c - C_i)\delta(\boldsymbol{y} - \boldsymbol{Y}_i)dB_i \tag{70}$$

and its quadratic variation is

$$
\begin{aligned}
&\nabla\nabla' : \left(\rho_n(t, \boldsymbol{y}, c)\rho_n(t, \boldsymbol{y}', c')cc'A_2([f_n(t) - f], \boldsymbol{y}, \boldsymbol{y}')\right)dt \\
&+ \partial_c\partial_{c'}\left(\rho_n(t, \boldsymbol{y}, c)\rho_n(t, \boldsymbol{y}', c')A_0([f_n(t) - f], \boldsymbol{y}, \boldsymbol{y}')\right)dt \\
&+ \partial_c\nabla'\cdot\left(\rho_n(t, \boldsymbol{y}, c)\rho_n(t, \boldsymbol{y}', c')c'A_1([f_n(t) - f], \boldsymbol{y}', \boldsymbol{y})\right)dt \\
&+ \partial_{c'}\nabla\cdot\left(\rho_n(t, \boldsymbol{y}, c)\rho_n(t, \boldsymbol{y}', c')cA_1([f_n(t) - f], \boldsymbol{y}, \boldsymbol{y}')\right)dt.
\end{aligned} \tag{71}
$$

With this calculation we obtain Dean's equation for the empirical distribution of the stochastic gradient descent process

$$
\begin{aligned}
\partial_t\rho_n = &\ \nabla\cdot(c\nabla U([\rho_n], \boldsymbol{y})\rho_n) + \partial_c\left(U([\rho_n], \boldsymbol{y})\rho_n\right) \\
&+ \tfrac{1}{2}\theta\nabla\nabla : \left(\rho_n c^2 A_2([f_n(t) - f], \boldsymbol{y}, \boldsymbol{y})\right) + \tfrac{1}{2}\theta\partial_c^2\left(\rho_n A_0([f_n(t) - f], \boldsymbol{y}, \boldsymbol{y})\right) \\
&+ \theta\partial_c\nabla\cdot\left(\rho_n c A_1([f_n(t) - f], \boldsymbol{y}, \boldsymbol{y})\right) \\
&+ \sqrt{\theta}\,\dot{\eta}_n(t, \boldsymbol{y}, c)
\end{aligned} \tag{72}
$$

where $f_n(t)$ is given by (5), i.e. $f_n(t, \boldsymbol{x}) = \int_{D \times \mathbb{R}} c\varphi(\boldsymbol{x}, \boldsymbol{y})\rho_0(t, \boldsymbol{y}, c)d\boldsymbol{y}dc$, and we defined the white-noise process $\dot{\eta}_n(t, \boldsymbol{y}, c)$ with quadratic variation in (71). The terms proportional to $\theta$ we refer to as $\mathcal{D}([\rho], \boldsymbol{y}, \boldsymbol{y})$ in the main text.

The deterministic part of (72) is the same as the equation for $\rho_0$ derived from GD. However, we have freedom as to the choice of $\theta$ so we can ensure that the noise terms occur at higher order, meaning that we recover the LLN. Specifically, we let

$$\theta = an^{-2\alpha} \quad \text{for some} \quad a > 0 \quad \text{and} \quad \alpha > 0 \tag{73}$$

This scaling can be achieved e.g. by choosing $P = O(n^{2\alpha})$, i.e. by increasing the batch size with $n$.

If $\alpha \in (0, 1)$ the fluctuations due to the noise in (72) eventually dominate the $O(n^{-1})$ fluctuations asymptotically. In this case, we have

$$\begin{aligned} \partial_t \rho_\alpha = \nabla \cdot & \left( -c\nabla F \rho_\alpha + \int_{D \times \mathbb{R}} cc' \nabla K(\boldsymbol{y}, \boldsymbol{y}') \left( \rho'_\alpha \rho_0 + \rho'_0 \rho_\alpha \right) d\boldsymbol{y}'dc' \right) \\ & + \partial_c \left( -F\rho_\alpha + \int_{D \times \mathbb{R}} c' K(\boldsymbol{y}, \boldsymbol{y}') \left( \rho'_\alpha \rho_0 + \rho'_0 \rho_\alpha \right) d\boldsymbol{y}'dc' \right) \\ & + \sqrt{a}\, \dot{\eta}_0(t, \boldsymbol{y}, c) \end{aligned} \tag{74}$$

in which $\dot{\eta}_0(t, \boldsymbol{y}, c)$ is a white-noise process with quadratic variation as in (71) but with $\rho_n$ replaced by $\rho_0$ and $f_n$ replaced by $f_0$.

Alternatively, if $\alpha \geq 1$, then the fluctuations due to the noise in (72) are negligible compared to the intrinsic flucutations, and we return to the setting of GD.

We hence take $\alpha \in (0, 1)$, and write both (74) as

$$\begin{aligned} \partial_t \rho_\alpha = \nabla \cdot & \left( c \int_\Omega \nabla_{\boldsymbol{y}} \varphi(\boldsymbol{x}, \boldsymbol{y}) \left( f_0(t, \boldsymbol{x}) - f(\boldsymbol{x}) \right) d\mu(\boldsymbol{x}) \rho_\alpha \right) \\ & + \partial_c \left( \int_\Omega \varphi(\boldsymbol{x}, \boldsymbol{y}) \left( f_0(t, \boldsymbol{x}) - f(\boldsymbol{x}) \right) d\mu(\boldsymbol{x}) \rho_\alpha \right) \\ & + \nabla \cdot \left( c \int_\Omega \nabla_{\boldsymbol{y}} \varphi(\boldsymbol{x}, \boldsymbol{y}) f_\alpha(t, \boldsymbol{x}) d\mu(\boldsymbol{x}) \rho_0 \right) + \partial_c \left( \int_\Omega \varphi(\boldsymbol{x}, \boldsymbol{y}) f_\alpha(t, \boldsymbol{x}) d\mu(\boldsymbol{x}) \rho_0 \right) \\ & + \sqrt{a}\, \dot{\eta}_0(t, \boldsymbol{y}, c) \end{aligned} \tag{75}$$

where we defined

$$f_\alpha(t, \boldsymbol{x}) = \int_{D \times \mathbb{R}} c\varphi(\boldsymbol{x}, \boldsymbol{y})\rho_\alpha(t, \boldsymbol{y}, c)d\boldsymbol{y}dc. \tag{76}$$

This equation is structurally similar to (48) except that it also contains a noise term. By proceeding similarly as we did to derive (51) we get the following equation for $f_\alpha(t, \boldsymbol{x})$:

$$\begin{aligned} \partial_t f_\alpha = & -\int_\Omega M([\rho_0(t)], \boldsymbol{x}, \boldsymbol{x}') f_\alpha(t, \boldsymbol{x}') d\mu(\boldsymbol{x}') \\ & - \int_\Omega M([\rho_\alpha(t)], \boldsymbol{x}, \boldsymbol{x}') \left( f_0(t, \boldsymbol{x}') - f(\boldsymbol{x}') \right) d\mu(\boldsymbol{x}') + \sqrt{a}\dot{\eta}(t, \boldsymbol{x}) \end{aligned} \tag{77}$$

where $M([\rho], \boldsymbol{x}, \boldsymbol{x}')$ is given in (19), and the quadratic variation of $\dot{\eta}(t, \boldsymbol{x})$ is precisely that of

$$\int_{D \times \mathbb{R}} c\varphi(\boldsymbol{x}, \boldsymbol{y})\dot{\eta}_0(t, \boldsymbol{y}, c)d\boldsymbol{y}dc \tag{78}$$

and given by

$$\begin{aligned} \langle d\eta(t, \boldsymbol{x}), d\eta(t, \boldsymbol{x}') \rangle = & \int_\Omega N([\rho_0(t)], \boldsymbol{x}, \boldsymbol{x}', \bar{\boldsymbol{x}}, \bar{\boldsymbol{x}}) \left| f_0(t, \bar{\boldsymbol{x}}) - f(\bar{\boldsymbol{x}}) \right|^2 d\mu(\bar{\boldsymbol{x}})dt \\ & - \int_{\Omega^2} N([\rho_0(t)], \boldsymbol{x}, \boldsymbol{x}', \bar{\boldsymbol{x}}, \bar{\boldsymbol{x}}') \left( f_0(t, \bar{\boldsymbol{x}}) - f(\bar{\boldsymbol{x}}) \right) \left( f_0(t, \bar{\boldsymbol{x}}') - f(\bar{\boldsymbol{x}}') \right) d\mu(\bar{\boldsymbol{x}})d\mu(\bar{\boldsymbol{x}}')dt \end{aligned} \tag{79}$$

in which

$$N([\rho], \boldsymbol{x}, \boldsymbol{x}', \bar{\boldsymbol{x}}, \bar{\boldsymbol{x}}') = \int_{(D \times \mathbb{R}^2)} \rho(\boldsymbol{y}, c) \rho(\boldsymbol{y}', c') \left( c^2 \nabla_{\boldsymbol{y}} \varphi(\boldsymbol{x}, \boldsymbol{y}) \cdot \nabla_{\boldsymbol{y}} \varphi(\bar{\boldsymbol{x}}, \boldsymbol{y}) + \varphi(\boldsymbol{x}, \boldsymbol{y}) \varphi(\bar{\boldsymbol{x}}, \boldsymbol{y}) \right)$$
$$\times \left( c'^2 \nabla_{\boldsymbol{y}'} \varphi(\boldsymbol{x}', \boldsymbol{y}') \cdot \nabla_{\boldsymbol{y}'} \varphi(\bar{\boldsymbol{x}}', \boldsymbol{y}') + \varphi(\boldsymbol{x}', \boldsymbol{y}') \varphi(\bar{\boldsymbol{x}}', \boldsymbol{y}') \right) d\boldsymbol{y} dc d\boldsymbol{y}' dc' \tag{80}$$

The SDE (77) has the property that it *self-quenches* as $t \to \infty$, because

$$\lim_{t \to \infty} A_k([f_0(t) - f], \boldsymbol{y}, \boldsymbol{y}') = 0, \qquad k = 0, 1, 2. \tag{81}$$

and we know that $f_0(t) \to f$ and from (81) we see that $\dot{\eta}(t) \to 0$ as well. Therefore, at long times (77) reduces to

$$\partial_t f_\alpha = - \int_\Omega M([\rho_0(t)], \boldsymbol{x}, \boldsymbol{x}') f_\alpha(t, \boldsymbol{x}') d\mu(\boldsymbol{x}') \tag{82}$$

Since $M([\rho_0(t)], \boldsymbol{x}, \boldsymbol{x}')$ is positive-definite the only (stable) fixed point of this equation is zero and $f_\alpha(t) \to 0$ as $t \to \infty$. Importantly, we need to choose large enough times that the fluctuations from the initial data have decayed. In this case, the LLN Proposition 1.2 still holds if we use the solution of (77) in (5), up to discretization errors.

## 5 Stochastic Gradient Descent: Asymptotic error

The considerations above also allow us to state:

**Proposition 5.1 (Asymptotic error for SGD)** *Let* $f_n(t) = f_n(t, \boldsymbol{x})$ *be as in* (5) *with* $\{\boldsymbol{Y}_i(t), C_i(t)\}_{i=1}^n$ *solution to* (67) *with* $\theta = an^{-2\alpha}$, $a > 0$ $\alpha \in (0, 1)$. *Then for any* $a_n > 0$ *such that* $a_n / \log n \to \infty$ *as* $n \to \infty$, *we have*

$$\lim_{n \to \infty} n^\alpha (f_n(a_n) - f) = 0 \qquad \text{almost surely} \tag{83}$$

In this statement, the almost sure convergence is with respect to $\mathbb{P}_{\text{in}}$ as well as the statistics of the noise terms in (67). In terms of the loss function, we have

$$\ell(f, f_n(a_n)) = \tfrac{1}{2} \|f - f_0(a_n)\|^2 - n^{-\alpha} \langle f - f_0(a_n), f_\alpha(a_n) \rangle + \tfrac{1}{2} n^{-2\alpha} \|f_\alpha(a_n)\|^2 + o(n^{-\alpha}) \tag{84}$$

and as a result we have

**Proposition 5.2** *Under the same conditions as those in Proposition 5.1, the loss function satisfies*

$$\lim_{n \to \infty} n^\alpha \ell(f, f_n(a_n)) = 0 \qquad \text{almost surely}. \tag{85}$$