[Reviews · NeurIPS 2018]

Reviewer 1



This paper adopts a perspective that has become popular recently, which views parameters in an optimization problem as defining measures in the scaling limit. The benefit of this approach is that non-convex optimization with the squared loss becomes a much simpler looking convex optimization problem over the space of measures. This paper uses this approach to derive based limit theorems for the large parameter regime. I am not an expert in this area, so while I find the perspective and results interesting and worth highlighting, I found the paper difficult to parse in places. I believe significantly more could be done to elucidate the content of the results, propositions 3.2 and 4.2 in particular. I will list some confusions I had in the course of reading the paper here, in the hopes that they will help the authors revise their work for a wider audience. The role of convexity: After (10), the authors note that the objective function becomes convex in the limit. But what is written in (10) is already a convex function on the space of measures, which has linear structure. Is what is gained in the limit the fact that the measure possesses a density with respect to the Lebesgue measure? (Is that why the authors emphasize the term density?) Or is some other base measure intended? Self-quenching for SGD: the authors highlight the particular self-quenching property of SGD, and argue in the introduction that this gives theoretical justification to the common belief that adding noise improves generalization. However, examining Prop 3.2 and its proof, I observe what I understood the authors to mean by "self-quenching" present in this setting as well—in the n \to \infty limit, the fluctuations of order n^(-1/2) grow smaller in the limit, so that in fact no fluctuations remain at order n^(-\xi) for \xi < 1. Is this indeed what is called self-quenching? (I.e., does it present the same phenomenon as is highlighted in Prop 4.2 and in the introduction?) If so, I am confused why this is highlighted as a virtue of adding noise, since the purely deterministic dynamics of GD also evince this behavior. Numerical experiments: These are slightly hard to interpret. First, which plots show SGD dynamics, and which are for GD? Second, I'm puzzled by how to interpret the dotted lines in each plot. In the case of RBF, how are we to make sense of the empirical n^{-2} decay? Is this somehow predicted in the analysis of the GD, or is it an empirical phenomenon which is not theoretically addressed in this work. Please clarify. In general, I do not find that the numerical experiments clearly support the story presented earlier in the work, except with respect to the broad qualitative fact of convergence in the n -> infty limit. ADDED AFTER FEEDBACK: I thank the authors for their response. As they revise this work, I encourage the authors to expand their discussion of self-quenching for SGD and clarify the experimental results. I would also encourage them to further discuss the connection to Dean's equation (the phrase "McKean-Vlasov equation" does not appear in the initial submission; this should be added if necessary). This would go a long way to making the methods used in the proofs clearer to the NIPS audience.

Reviewer 2



This article maps weights of a neural net that are being learned to particles, and views the learning procedure and dynamics of the particles. The article derives in an algorithmically constructive way the universal approximation theorem, and scaling of the error with the number of hidden units. The results are then demonstrated on learning of the energy function of the p-spin model. It is worth noting that the conclusions of the article are insensitive to details of the network architecture. The general direction this papers takes is quite interesting. However, the paper is not very clear in what is actually the main contribution and hos is it related to existing work. I note some potentially questionable points below: ** The abstract states: "From our insights, we extract several practical guidelines for large scale applications of neural networks, emphasizing the importance of both noise and quenching, in particular." It was not really clear to me from the paper what are these "practical guidelines". ** The article analyzes online gradient descent, where a new mini-batch is sampled every iteration and is never reused in the learning afterwards. While this is an interesting learning dynamics that attracted attention, this is not the same as SGD that usually reuses samples. This assumption seems well hidden in the paper on line 163. Instead it should be stated from the beginning in order not to confuse the reader. ** Maybe a word or two could be spend on the sample complexity of the procedure in the paper. It seems that we need very many samples, but how many? Do we know how does their number and the training time scale with n and d? ** The analysis of the dynamics that maps weights on particules and described then the dynamics via an analog of the Dean's equation (relatively well known in statistical physics) is interesting and will be particularly appealing to physics audience. ** The work of F. Bach "Breaking the curse of dimensionality with convex neural networks." seems to be related to the present paper, especially concerning the result about convexity of learning when the hidden layer is large. However, this relation (and novelty wrt that work) is not discussed in the present paper. ** I found that the example of the p-spin model on which the authors demonstrate their results in not very well motivated. I see that it is a complicated function to fit, but does learning of this function arise in some applications? ** The plot in Fig. 1 should be better explained. In the first pannel the authors plot 1/n and 1/n^2, but it is not clear why the 1/n is a better fit than the other. The data seems rather noisy and only about one decade is plotted. Should this be considered as a convincing evidence for the derived scaling? Page 8 refers to Fig. 5, should this be Fig 1? ADDED AFTER FEEDBACK: My comments n. 3,4,5 were well taken into account in the rebuttal. As for n. 1: Scaling in n is still not a practical guideline since usually one deals with a system of a given size and one does not have any access to scaling. Thus I would recommend the claims about practical guidelines to be tuned down. This paper is far away from any *practical guidelines*. As for n. 2: This comment about SGD versus online was ignored in the rebuttal, while I think this assumption is strong and should be stressed. As for n. 6 & 7: The rebuttal repeats what is already written in the paper. To me the interest of this particular example as well as the presentation of the numerical results remains obscure. Overall while the presentation of this paper is far from ideal, yet the approach is interesting as well as the theoretical results. The above drawbacks prevent me from raising the score to more than 5, but I decided to raise to 5.

Reviewer 3



In this manuscript the authors show that neural networks can be used to get approximations to given high-dimensional function with controllable errors. In particular, they consider the case where the function parameters satisfy either Gradient Descent or Stochastic Gradient Descent dynamics and prove law of large numbers and central limit theorems for the function the neural network outputs. They then use simulations, where the input function is a spherical 3-sping model, to show the performance of neural networks in estimating these functions. I find the paper to be very clear and well-written. The authors do a good job in setting up the problem, discussing the two different settings and explaining the results. Those results are in my opinion very relevant since they shed new light, and provide important insights, in the strength of neural networks for representing complex high-dimensional functions, which can have several important applications in both theory and practice. The authors have chosen to put all technical derivations and details in the supplementary material. Given the depth of the paper I understand this choice. However, the links between the parts of the main text and the supplementary material are not very clear. For instance, the authors use S1 in the main text to refer to Section 1 in the supplementary material, although sections there are simply numbered by 1, 2 and so on. Moreover, I found it hard to move between the main and supplementary parts in a continuous way. I ended up first reading the main text and then the supplementary material, which made it very hard to directly related the results in the latter to the main text. Here the problem arises that not everything used in the supplementary material is defined there, forcing the reader to go back to the main text and finding the missing definitions. Finally, I want to mention that the supplementary material does not follow the standard setup of first given a statement and then a detailed proof of it. Instead the results are preceded by lengthy computations which should represent the proof of this statement. This works in some case, but in most I was unable to parse the derivations I just read to get the proof of the result. I therefore stopped reviewing the computations after page 6 because of the difficulties in parsing them into needed proofs. To summarize, the paper investigates the very interesting and important usage of neural networks for representing complex high-dimensional functions using input parameters coming from given dynamics. The results seem solid and establishing the relation between function representation by neural networks and particle dynamics can potentially lead to better understanding and new advances of these systems. However, the way in which the results and technical derivations are split between the main text and supplementary material make it very difficult to properly check and verify the results. Still, I would recommend this paper for acceptance in NIPS, given that the authors take some time to better structure their results. ADDED AFTER FEEDBACK: I remain convinced that the approach in the paper and results are interesting enough to validate acceptance in NIPS 2018. Although I strongly encourage the authors to restructure their supplementary material and discuss the contributions of the paper more clearly. Below is a detail list of comments. Main text: Line 14, [batch size]: It is not clear to the reader at this point what this means. Line 31, [Progress analyzing…]: I would write: Progress on analyzing. Line 56: Another instance where batch size is used without having explained what is means. Line 59: I was confused by: justification explanation. It feels that it should be just one of these words. In addition, I would remove [in] in: noise in can lead… Line 70, [an exacting test]: Don’t you just mean: a test? Line 74-75: I would remove: is the following criterion, since it repeats the part of the sentence that precedes it. Equations (4) and (5): Here the use of h is confusing since it was just use to denote an arbitrary test function. Line 86, [In this case, we…]: I would write: In this case, since we… Also, I would remove [and] in: and we can view… Line 92, [converges]: What type of convergence? I guess you mean pointwise. Line 104, [synergies will…]: I would write: synergies which will… Line 114: As mentioned before, there is no Section S1. Moreover, it really is Section 1.1 in the supplementary material. I would recommend to be more specific here to avoid confusion and unnecessary text scanning by the reader. This holds for all subsequent occurrences of S references. Equation (15): It is unclear with respect to what variables the integrals are taken. Please be explicit here. Equation (16): What is \rho_0? It is never defined and clearly cannot be a case of eq (13). Could you please give some more details here. Line 121: I would remove [and] in: point is and… Line 125: What is the difference between f_n(t) and f_n(x)? This is not clear from the text. Line 127: The \mathbb{P}_{in} comes out of nowhere which creates confusion. I would advise to spend some words on this before the statement of Proposition 3.1. Equation (19): Is this convergence uniform in t? I was not able to extract this from the proof and it feels that this is somewhat necessary to make the result applicable. This also applies to eq (29). Line 160: Here you refer to equation (27) which comes only later. I would recommend to follow the practice of not referring ahead and first give (27). In addition, I would write: (27) is statistically… Line 162, [spation-temporal]: I am not completely familiar with the terminology in Langevin dynamics but I would think that spatial-temporal is the right term. Line 172: What are \bm{\eta} and \eta? I understand that they refer to noise terms but it would be better to at least refer to the equation in the supplementary material that defines them. Line 192, [of noise term…]: I would write: of the noise term… Line 205: It would help the reader if you explain what S^{d-1}(\sqrt{d}) is. Figure 1: This figure is unclear. For instance, what do the different color stars mean? I would strongly recommend to add more explanation in the caption. Also, the figure is never explicitly referenced in the main text. Line 241, [that the]: I would replace this with: where the Supplementary material: Line 12: Are there really no restrictions on the input distribution \mathbb{P}_{in}? Line 14: What is \Omega? This is never defined. Line 37: What does the half arrow mean? I guess it means convergence in distribution. Please be more specific. Line 42-Equation (16): It is not clear why any of this is needed, since it does not seem to be used in the derivations that follow. Moreover, what is the derivative in eq (11)/(13)? Is this the Fréchet derivative? If so then why does \frac{\partial \mathcal{E}_0}{\partial \rho_0} seem to be a function, while \mathcal{E}_0 is a functional? It would really help if you give more details and a motivation for this part. Equation (18): I did not immediately see who (10) can be rewritten as (18). Could you please add some additional derivations? Line 58, [By the assumptions…]: Which assumptions are you referring to? The only assumption I could find here is 1.2 which says the kernel is continuously differentiable but I do not see why that is enough. Proposition 1.3: I was unable to transform the computations preceding this proposition into a proof of it. Please try and make the proof more apparent from the computations so that the reader does not need to solve this puzzle themselves. Line 72: I did not understand why the minimizers of \mathcal{E}_0 are relevant. Could you please elaborate a bit on this? Line 93: There seems to be a typo here. Along is written twice. Equation (32): I found the notation here quite confusing. First it is only revealed later that \xi(t) also depends on n which is indeed needed for else (32) would not be true in general. Second, what is the difference between \tilde{\rho}_\xi(t) and \tilde{\rho}_{\xi(t)}? Equation (38): What asymptotic are you considering here? Is it n \to \infty, t \to \infty or both n, t \to \infty?